# Strong intrinsic room-temperature ferromagnetism in freestanding non-van der Waals ultrathin 2D crystals

Hao Wu[1], Wenfeng Zhang[1], Li Yang[1], Jun Wang[2], Jie Li[1], Luying Li[3], Yihua Gao [3,4], Liang Zhang[5], Juan Du[6,7], Haibo Shu[2] & Haixin Chang [1,5,8 ✉]

Control of ferromagnetism is of critical importance for a variety of proposed spintronic and topological quantum technologies. Inducing long-range ferromagnetic order in ultrathin 2D crystals will provide more functional possibility to combine their unique electronic, optical and mechanical properties to develop new multifunctional coupled applications. Recently discovered intrinsic 2D ferromagnetic crystals such as $Cr_2Ge_2Te_6$, $CrI_3$ and $Fe_3GeTe_2$ are intrinsically ferromagnetic only below room temperature, mostly far below room temperature (Curie temperature, ~20–207 K). Here we develop a scalable method to prepare freestanding non-van der Waals ultrathin 2D crystals down to mono- and few unit cells (UC) and report unexpected strong, intrinsic, ambient-air-robust, room-temperature ferromagnetism with $T_C$ up to ~367 K in freestanding non-van der Waals 2D CrTe crystals. Freestanding 2D CrTe crystals show comparable or better ferromagnetic properties to widely-used Fe, Co, Ni and $BaFe_{12}O_{19}$, promising as new platforms for room-temperature intrinsically-ferromagnetic 2D crystals and integrated 2D devices.

[1] Center for Joining and Electronic Packaging, State Key Laboratory of Material Processing and Die & Mold Technology, School of Materials Science and Engineering, Huazhong University of Science and Technology (HUST), Wuhan 430074, China. [2] College of Optical and Electronic Technology, China Jiliang University, Hangzhou 310018, China. [3] Center for Nanoscale Characterization and Devices, Wuhan National Laboratory for Optoelectronics, Huazhong University of Science and Technology, Wuhan 430074, China. [4] School of Physics, Huazhong University of Science and Technology, Wuhan 430074, China. [5] School of Science and Center for Materials Science and Engineering, School of Microelectronics and Materials Engineering, Guangxi University of Science and Technology, Liuzhou, China. [6] Ningbo Institute of Material Technology & Engineering, Chinese Academy of Sciences, Ningbo 315201, China. [7] Institute of Materials, Shanghai University, Shanghai 200444, China. [8] Institute for Quantum Science and Engineering, Huazhong University of Science and Technology, Wuhan 430074, China. ✉email: hxchang@hust.edu.cn

Long-range ferromagnetic order in 2D isotropic systems is intrinsically forbidden at non-zero temperature according to the Mermin–Wagner theorem[1]. Intrinsic room-temperature 2D ferromagnetism in ultrathin two-dimensional (2D) crystals is strongly suppressed and rarely observed. Recent efforts to explore the ferromagnetism in ultrathin 2D crystals have revealed intrinsic ferromagnetism in mechanically exfoliated $Cr_2Ge_2Te_6$, $CrI_3$, and $Fe_3GeTe_2$ few-layer crystals, but the ferromagnetic ordering only exists significantly far below room temperature[2–9]. Moreover, most intrinsic ultrathin ferromagnetic 2D crystals have a van der Waals structure and are often exfoliated mechanically by tape[10–13]. No intrinsic room-temperature 2D ferromagnetism is found in freestanding, ultrathin, non-van der Waals 2D crystals, which consist of larger numbers of materials than van der Waals-type crystals, partly a result of the great difficulty in obtaining non-van der Waals 2D crystals with high magnetic anisotropy.

Transition-metal mono-chalcogenide CrTe has been demonstrated to be ferromagnetic in bulk with zinc blende (zb)[14,15] or NiAs[16,17] crystal structures. Only NiAs type shows room-temperature ferromagnetism with a magnetic anisotropy property and a Curie temperature ($T_C$) mostly ~340 K[16,17]. However, it is a big challenge to obtain ultrathin 2D crystals with maintained room-temperature ferromagnetism in such non-van der Waals crystals with specific crystal phase. Considering the difficulty to mechanically exfoliate non-van der Waals crystals using tape, here we develop a CVD-assisted ultrasonication method to obtain ultrathin freestanding non-van der Waals 2D CrTe crystals with thickness down to mono-unit cells (UC), few-UC, and multi-UC. In contrast to intrinsic, van der Waals ferromagnetic 2D crystals with much lower $T_C$ than bulk crystal[2–5], freestanding non-van der Waals 2D CrTe crystals have higher $T_C$, up to 3.7 times larger spontaneous magnetization ($B = 0$, magnetization regime without external magnetic field, or spontaneous spin polarization), 35% larger saturation magnetization, and 10.5 times larger coercivity than bulk CrTe crystals, which we attribute to a 2D quantum confinement-induced enhancement of spin polarization with reduction of dimension.

## Results

**Exfoliation of the non-van der Waals CrTe crystals.** The unit cell structure of CrTe is shown in Fig. 1a, and Cr and Te atoms are hexagonally distributed as viewed along [002] direction. The calculated cleavage energy of (002) face (37.87 meV/$Å^2$) of CrTe is the lowest compared with other faces (Fig. 1b–e) and is comparable to the cleavage energy of graphite (23.13–24.38 meV/$Å^2$)[18], indicating the possibility to large-scale ultrasonication-based exfoliation like most van der Waals crystals, as illustrated in Fig. 1f. The CrTe nanosheet single crystals are first grown on $SiO_2$/Si substrate by CVD process followed by ultrasonication in solvents (Supplementary Fig. 1). The CVD-grown CrTe crystal is NiAs type with a (002) face-preferred growth (Fig. 1g and Supplementary Fig. 2), which enables the exfoliation even by scotch tapes. By controlling the ultrasonication and settling time (see 'Methods'), freestanding multi-UC and few-UC CrTe single crystals with narrow thickness distributions can be obtained. The thickness of few-UC CrTe crystals can be even down to mono-UC, bi-UC, and tri-UC (Fig. 1h–j) (We note 0.8 nm corresponds to mono-UC considering a theoretical mono-UC of 0.624 nm.). The process of ultrasonication is completely a physical and mechanical process that does not change the crystal structure of CrTe. As shown in Supplementary Fig. 3a, the powder X-ray diffraction (XRD) spectra in few-UC and multi-UC CrTe crystals confirm the standard NiAs-type crystal phase with (002) face-preferred growth. The atomic ratios of Cr:Te in obtained multi-

UC and few-UC CrTe crystals are similar with bulk crystal and very close to 1:1 for typical stoichiometry of CrTe (Fig. 2c, inset and Supplementary Figs. 3b, 4d, and 5d).

**Characterization of freestanding few-UC and multi-UC CrTe crystals.** We dispersed both the few-UC and multi-UC CrTe crystals in ethanol solvent with a mass concentration of ~0.05–0.1 mg/ml, as shown in Fig. 2a and Supplementary Fig. 4a. The few-UC and multi-UC CrTe crystals well dispersed in ethanol can be attracted easily when a ferromagnet is put nearby at room temperature, and can be easily re-dispersed again, which is attributed to the non-van der Waals nature of the CrTe ultrathin 2D crystals that prefer to be freestanding rather than restacking again once exfoliated. The thickness of few-UC CrTe crystals is mostly between 2 and 4 nm (~2–5 UC, Fig. 2b), while that of multi-UC CrTe crystals is mainly between 20 and 40 nm (~20–50 UC, Supplementary Fig. 4b). Instead of breaking into nanoparticles after ultrasonication, the size of few-UC and multi-UC CrTe crystals is still maintained at the level of several to 10 μm, as shown in Fig. 2c and Supplementary Figs. 4c–e and 5. Element mapping of Cr and Te by energy-dispersive X-ray spectroscopy (EDX) on the few-UC CrTe crystal confirms the atomic homogeneity after ultrasonication (Fig. 2d), which further proves the physical, mechanical characteristics during the ultra-sonication procedure after CVD process. Figure 2e shows high-resolution transmission electron microscopy (HRTEM) image at the edge of the few-UC CrTe crystal. Clear lattice fringes can be seen when the CrTe layer is perpendicular to the direction of the electron beam, proving the high degree of crystallinity of few-UC CrTe. In addition, AFM images at the edges of exfoliated CrTe crystals confirm the nanosheet structures (Supplementary Fig. 6). Figure 2f shows the atomic-resolution TEM image of few-UC CrTe crystal. Clear (110) lattice planes can be identified, which shows an interplanar spacing of ~0.20 nm. Selected-area electron diffraction (SAED) (Fig. 2f and Supplementary Fig. 5c) exhibits two-fold hexagonal diffraction spots, indicating the (002) exposed face of few-UC and multi-UC CrTe and demonstrating the (002) cleavage plane of CrTe, consistent with previous calculations. Figure 2g shows the reduced fast Fourier transformation (RFFT) image derived from Fig. 2f, and same interplanar spacing of ~0.20 nm for (110) interplanar spacing is observed, consist with the atomic-resolution TEM result in Fig. 2f. Some disorders are found at the edge areas (Fig. 2e), which is frequently seen in ultrathin 2D crystals[19].

**Ferromagnetic properties of freestanding few-UC and multi-UC CrTe crystals.** To investigate the ferromagnetism of multi-UC and few-UC CrTe crystals, we performed vibrating sample magnetometer (VSM) measurements at varied temperatures. We use a millipore filter substrate to collect the crystals with the sheets parallel to filter substrate as shown in Supplementary Fig. 5a, b (details in 'Methods'), and the influence of background signals such as the millipore filter substrate is excluded (Supplementary Fig. 7). Figure 3a–c shows the moment–magnetic field (MH) curves of bulk, multi-UC, and few-UC CrTe crystals from 2 to 400 K and a field range of −5~5 T. Bulk CrTe shows typical ferromagnetic MH curves with $T_C$ around ~343 K, and typical saturation magnetization of 69.5 emu/g (~2.30 μB per Cr) at 2 K and 35.9 emu/g (1.16 μB per Cr) at 300 K, consistent with the reported ones[20]. Compared with bulk CrTe, the multi-UC CrTe crystals show a highly enhanced saturation magnetization up to 93.7 emu/g (3.10 μB per Cr) at low temperature (<100 K) and 48.5 emu/g (1.60 μB per Cr) at 300 K, which is attributed to 2D quantum confinement effect with reduction of dimension and resulted in electronic structure changes, especially spin

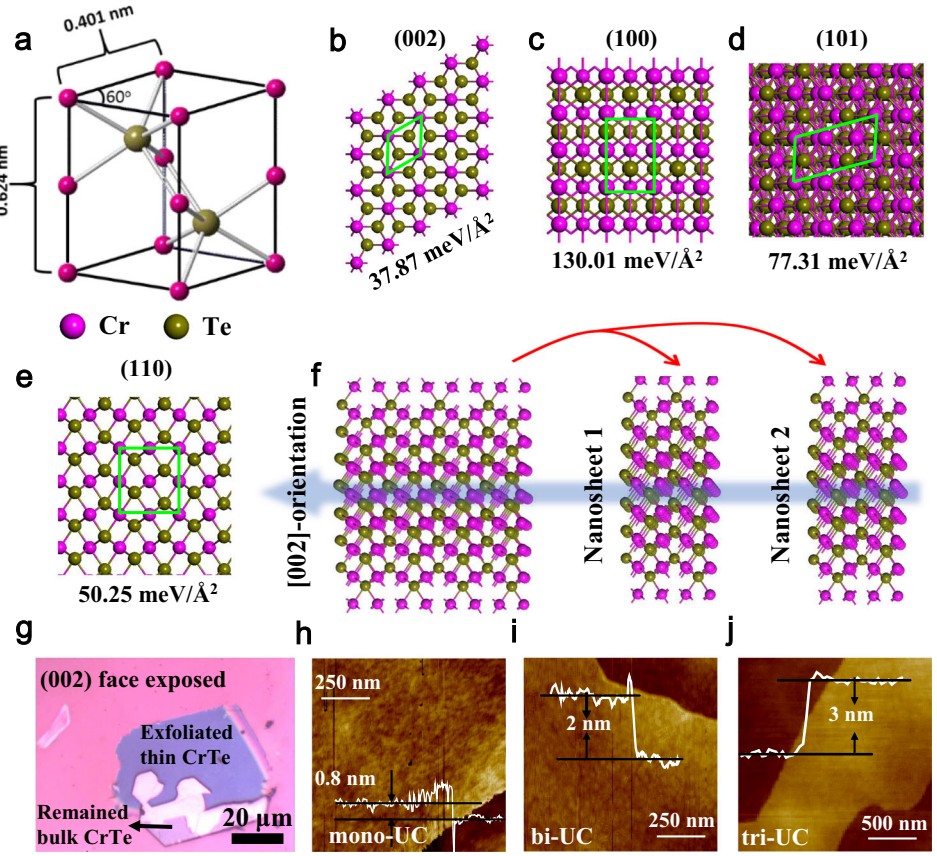

**Fig. 1 Exfoliation of the non-van der Waals CrTe crystals by CVD-assisted ultrasonication. a** Unit cell of NiAs-tape CrTe. The purple and brown balls represent Cr and Te atoms, respectively. **b–e** The surface structures and calculated exfoliation energies $E_{ex}$ (in meV/Å$^2$) of four CrTe crystal faces: **b** (002), 37.87 meV/Å$^2$; **c** (100), 130.01 meV/Å$^2$; **d** (101), 77.31 meV/Å$^2$; and **e** (110) 50.25 meV/Å$^2$. The green frames in (**b–e**) denote the range of unit cell. **f** Mechanism diagram of the cleavage direction for CrTe crystal. **g** A (002) face exposed bulk CrTe crystal on a SiO$_2$/Si substrate that been mechanically exfoliated easily by a Scotch tape, indicating a small cleavage energy of (002) face. **h–j** AFM images for mono-UC, bi-UC, and tri-UC 2D CrTe crystals obtained by CVD-assisted ultrasonication with different thicknesses of 0.8 nm (mono-UC), 2 nm (bi-UC), and 3 nm (tri-UC).

polarization enhancement (see discussion about first-principles calculations below). The ferromagnetic hysteresis loop can hold at a higher temperature over 360 K in multi-UC CrTe (Supplementary Fig. 8e). The saturation magnetization of few-UC CrTe is up to 89.2 emu/g (2.94 μB per Cr) at 2 K and 19.7 emu/g (0.67 μB per Cr) at 300 K, which is smaller than that of multi-UC CrTe crystals. The smaller magnetic moment may result from the defect in edge areas of few-UC CrTe as shown in TEM imaging (Fig. 2e) and XRD spectra (Supplementary Fig. 3a), and theoretical discussions below. The hysteresis loop in few-UC CrTe preserves even at 400 K (Fig. 3c and Supplementary Fig. 8f). The in-plane and out-of-plane magnetization at room temperature show an in-plane easy magnetization axis in both multi-UC and few-UC CrTe crystals (Supplementary Fig. 9), agreeing with previous work[21].

To further explore the influence of the thickness on magnetism of ultrathin CrTe crystals, temperature-dependent tests under spontaneous magnetization regime without magnetic field ($B = 0$, or spontaneous spin polarization, more details in 'Methods') and magnetization under 0.3 T zero-field-cooling (ZFC) and field-cooling (FC) tests are further performed on bulk, multi-UC, and few-UC CrTe crystals (Fig. 3d, e). Spontaneous magnetization ($B = 0$) for few-UC and multi-UC crystals are ~4.7 and ~2.3 times that of bulk CrTe crystal at 2 K, respectively (Fig. 3d). The $T_C$ increases from ~343 K for bulk to ~367 K for multi-UC CrTe (Fig. 3d, e and Supplementary Fig. 10), which is consistent with MH results (Supplementary Fig. 8). This is in contrast to the van

der Waals crystals where Currie temperature decreases with the thickness reduction and van der Waals ultrathin crystals have much lower $T_C$ than bulk[2–5]. As for few-UC CrTe, two kinds of room-temperature ferromagnetism are observed, strong intrinsic ferromagnetism with $T_C \sim 350$ K (Supplementary Fig. 10) and very weak ferromagnetism (~400 times smaller saturated magnetization than intrinsic ferromagnetism) over 350 K probably from defects at edge areas (Fig. 3d, e and Supplementary Fig. 8f)[22,23]. The coercivity of few-UC CrTe crystals is ~1.5 and ~10.5 times larger than multi-UC ones and bulk CrTe crystals at 2 K, respectively (Fig. 3f). At room temperature, the coercivity of few-UC CrTe crystals is slightly higher than multi-UC ones and is still ~5.0 times larger than bulk CrTe crystals. Importantly, both multi-UC and few-UC CrTe crystals show comparable or better saturation magnetic moment and coercivity when compared with conventional ferromagnetic materials such as Fe, Co, Ni, and BaFe$_{12}$O$_{19}$ (a material widely used in high-density magnetic recording media and permanent magnet) (Fig. 3g–i and Supplementary Table 1, more discussions in notes for Supplementary Table 1)[24]. These results indicate that freestanding non-van der Waals 2D crystals provide a new route to obtain and tune the intrinsic room-temperature 2D ferromagnetism in 2D crystals, making it easier and more possible to achieve very different physical properties from traditional ultrathin films on substrates (see more in Supplementary Discussion 1 in Supplementary information)[25]. We also note that the observed beyond-room-temperature 2D ferromagnetism is intrinsic without any

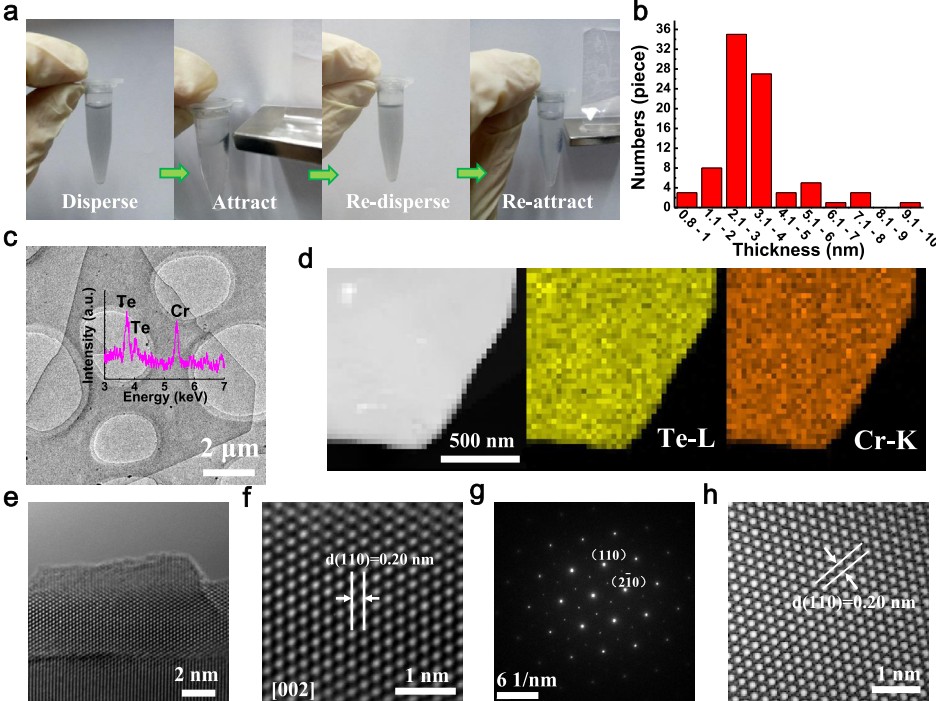

**Fig. 2 Characterization of freestanding few-UC 2D CrTe crystals. a** Freestanding few-UC CrTe crystals are well dispersed in ethanol, attracted by a ferromagnet and well re-dispersed after attraction. **b** Thickness distribution of the freestanding few-UC CrTe crystals from AFM statistical analysis. **c** Micrometer-scale TEM image of the few-UC CrTe crystals on milliporous substrate. **d** Dark-field image and corresponding element mappings for the few-UC CrTe crystal. **e** HRTEM image at the edge of the few-UC CrTe crystal. **f** Atomic-resolution TEM image of the few-UC CrTe crystal. **g** The selected-area electron diffraction (SAED) for the few-UC CrTe crystal. **h** Reduced fast Fourier transformation (RFFT) image derived from (**g**).

requirement of extrinsic modulations such as external electric field stimulation or extra support of interacting substrates like in van der Waals $Fe_3GeTe_2$ and $VSe_2$ 2D crystals[26,27].

**Micro-area ferromagnetic properties and the mechanism of the thickness-dependent ferromagnetism.** To further characterize the thickness-dependent ferromagnetism in CrTe crystal, anomalous Hall effect (AHE), magneto-optical Kerr effect (MOKE), as well as magnetic force microscopy (MFM) measurements were performed on single sheet CrTe crystals with different thickness. For AHE tests, large area as-grown CrTe crystals and five-wire Hall device were employed, as shown in Supplementary Fig. 11. Supplementary Figs. 12 and 13 show the temperature-dependent Hall resistivity for four different thickness CrTe crystals (100, 69, 37 nm thin crystals and 20 μm bulk crystal). The Curie temperatures of 100, 69, and 37 nm CrTe crystals are ~340, ~370, and ~360 K, respectively, all greater than the 330 K Curie temperature of bulk CrTe. The Curie temperature increases first and then decreases with the decrease of thickness, which is consistent with the previous VSM results. At room temperature, the anomalous Hall resistivities of bulk, 100, 69, and 37 nm CrTe crystals are 0.64, 0.82, 12.7, and 1.95 μΩ·cm, respectively. The anomalous Hall resistivity also increases first and then decreases with the decrease of thickness, which is consistent with the change trend of $T_C$. Single sheet MOKE measurements also show a higher $T_C$ in single sheet multi-UC CrTe crystals than in single sheet bulk CrTe crystals (Supplementary Fig. 14a–c, more discussions in notes for Supplementary Fig. 14), consistent with the VSM and AHE tests. In addition, a room-temperature magnetic linear domain wall (LDW) magnetic domain behavior is observed in single sheet multi-UC CrTe crystals (Supplementary Fig. 14d–g), quite different from the random magnetic domain walls in bulk CrTe crystals (Supplementary Fig. 15).

Ferromagnetic properties of different thickness single sheet CrTe at nanoscale are further directly imaged by MFM at room temperature without external magnetic field applied on the samples (Supplementary Fig. 16 and Fig. 4). When the CrTe crystal is in bulk form (20 μm), the shape of the magnetic domain wall is arranged randomly, as shown in Supplementary Fig. 16a, b. When the thickness of CrTe is reduced to several hundreds of nanometers (673 nm), the shape of the magnetic domains is a mixture of random and striped magnetic domains (Supplementary Fig. 16c–e). When the thickness is reduced to 96, 39, and 34 nm, only linear magnetic domains are found (Supplementary Fig. 16f, g). The thickness-dependent domain wall behavior measured by MFM is consistent with the MOKE results. When the thickness is further reduced to several nanometers, there is only one magnetic domain on each single sheet, whose shape is exactly the same as the CrTe crystal (Fig. 4). To avoid the contribution of short-range atomic force influence when domain shape is the same as the crystal shape, the pre-magnetized AFM tip is set at 50 nm away from the sample (Supplementary Fig. 17), and a MFM phase image along with AFM topography is also obtained for an ~8 nm thick (~10 UC) CrTe (Fig. 4a) (more discussion in notes for Supplementary Fig. 17). Figure 4b shows the AFM images and corresponding in-site MFM phase images for single or stacked ~3 nm thick (~tri-UC) CrTe nanosheet, which exhibits uniform MFM phase signals through each crystal. Figure 4c–e shows the MFM phase signals of the tri-UC and stacked tri-UC CrTe crystals with different selected typical side width. The difference in MFM phase signals for bare substrate, single tri-UC sheet, and two stacked tri-UC sheets can be distinguished even when the selected typical side width is down to 10 nm scale.

To gain more insight into the nature of the thickness dependence of magnetism in ultrathin CrTe nanosheet crystals, we studied the electronic structures of bulk, multi-UC, few-UC,

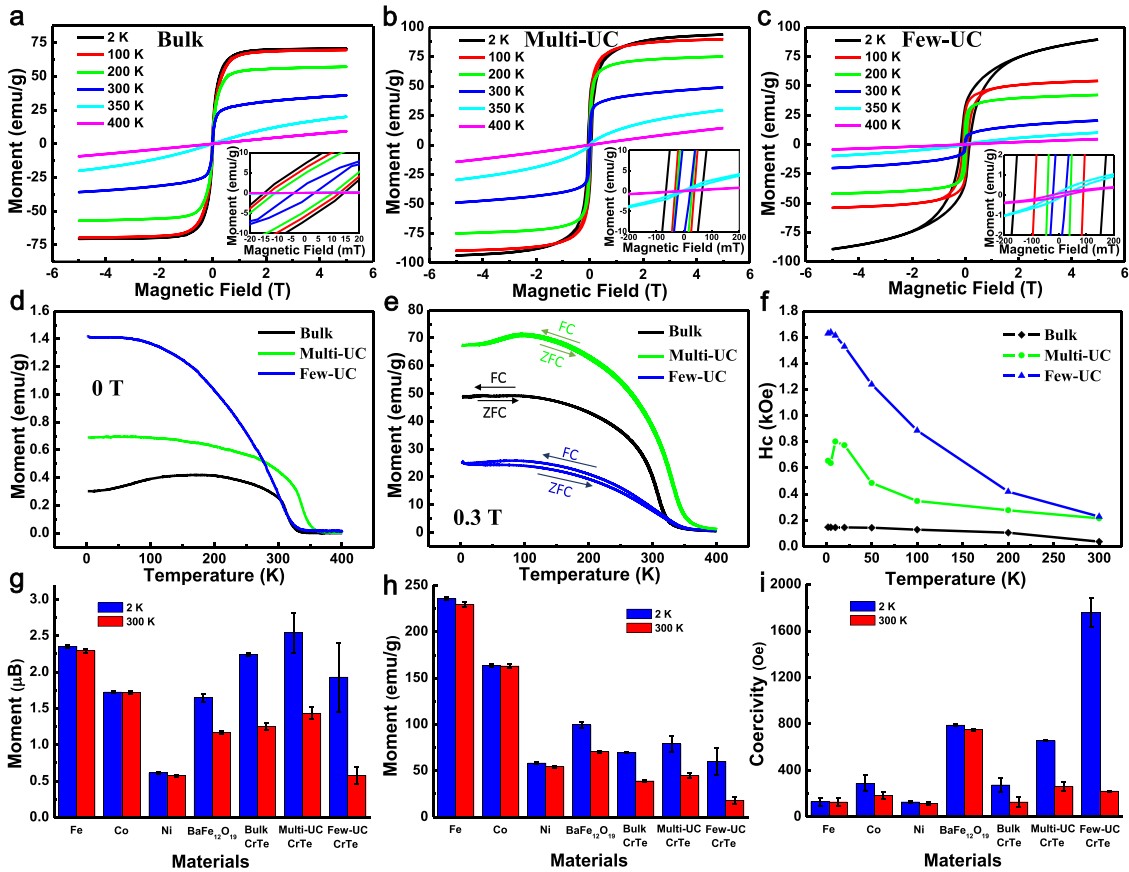

**Fig. 3 Ferromagnetic properties of bulk, multi-UC, and few-UC CrTe crystals. a–c** M–H hysteresis of bulk, multi-UC, and few-UC CrTe crystals from 2 to 400 K. **d** Spontaneous magnetization without magnetic field ($B = 0$) for bulk, multi-UC, and few-UC CrTe crystals from 2 to 400 K. **e** Temperature dependences of moment (M–T) for bulk, multi-UC, and few-UC CrTe from 400 to 2 K with 0.3 T magnetic field using ZFC and FC tests. **f** Temperature dependences of coercivity (Hc) for bulk, multi-UC, and few-UC CrTe crystals from 2 to 300 K. **g–i** Saturation magnetic moment (**g**, **h**) and coercivity (**i**) comparison of bulk, multi-UC, and few-UC CrTe crystals with Fe, Co, Ni, and $BaFe_{12}O_{19}$. Error bars s.e.m., $N = 3$; see original data in Supplementary Table 1 and notes in Supplementary information.

and disordered few-UC and mono-UC CrTe by using density functional theory (DFT)-based first-principles calculations (Supplementary Figs. 18–21). The calculated spin-resolved density of states (DOSs) and band structures indicate that the majority-spin and minority-spin electronic states in all the CrTe crystals are asymmetric, especially for electronic states around Fermi level (see Supplementary Figs. 18–20), which is responsible for and consistent with the observed ferromagnetic behavior in bulk, multi-UC, and few-UC CrTe while direct mono-UC CrTe ferromagnetic test is beyond instrument sensitivity like many monolayer cases in van der Waals 2D crystals (more discussion in notes for Supplementary Fig. 20)[2]. The calculations indicate that bulk CrTe crystal is metallic, consistent with the reported ones (Supplementary Figs. 18a and 19a)[20], and show significantly increased even half-metallic spin polarizations with reduction of thickness and dimension due to the 2D quantum confinement effect (Supplementary Figs. 18b, c and 19b, c). The analysis of DOSs and spin density also shows that the magnetic moments are mainly contributed by the Cr-3$d$ electrons, and each Cr atom carries increased average magnetic moment of 3.95, 4.13, and 4.29 $\mu_B$ for bulk, multi-UC, and few-UC CrTe (Supplementary Fig. 21), respectively. The enhanced spin polarizations in 2D CrTe crystals than in bulk CrTe crystals from calculations are also consistent with the spontaneous magnetization ($B = 0$) tests, where spontaneous magnetization and spontaneous spin polarization in few-UC and multi-UC CrTe crystals is ~4.7 and ~2.3

times that of bulk CrTe crystal, respectively (Fig. 3d), considering that the spontaneous magnetization ($B = 0$) is highly related with the spontaneous spin polarization of the carriers or conduction electrons[9,28]. These calculations are also consistent with the observed higher $T_C$ and larger saturation magnetization in multi-UC 2D CrTe crystals than that in bulk CrTe crystals. However, as mentioned above, the measured saturation magnetization and $T_C$ of few-UC CrTe is lower than that of multi-UC, which may arise from the observed disorder in few-UC CrTe (Fig. 2h). The calculations show that severe structural disorder may lead to changes from typical Cr–Cr ferromagnetic coupling to both Cr–Cr ferromagnetic and antiferromagnetic couplings (Supplementary Fig. 18e, f), resulting in dramatic degeneration of spin polarization and decrease of average magnetic moment per Cr atom from 4.29 to 0.84 $\mu_B$ (Supplementary Fig. 18c, d). Therefore, the results here may provide a new route to tune and achieve high spin polarization in 2D ferromagnetic crystals and may make new quantum devices possible based on spontaneous spin polarization in 2D crystals (more discussions in Supplementary Discussion 2 in Supplementary information).

Most of the discovered freestanding van der Waals ultrathin 2D ferromagnetic crystals show unavoidable instability in long-term exposure of ambient air atmosphere, which limit their practical applications. As an inspiring result, the multi-UC and few-UC CrTe crystals show unusual crystal and ferromagnetism stability in long term, ambient air conditions (Supplementary

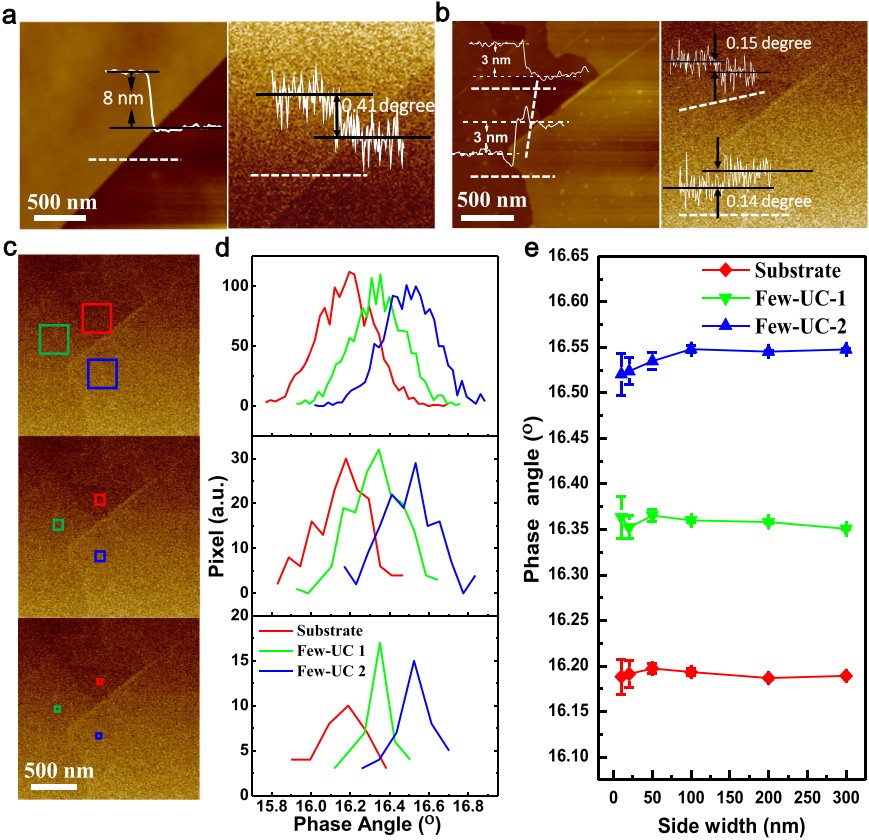

**Fig. 4 Direct imaging of the room-temperature ferromagnetic properties of single sheet few-UC CrTe crystals by magnetic force microscopy (MFM) at nanoscale. a** Topography image (left) of an 8 nm thick (~10 UC) CrTe crystal with corresponding in-site MFM phase image (right) by a pre-magnetized AFM tip. **b** Topography images (left) with corresponding in-site MFM phase images (right) for several stacked 3 nm thick (~tri-UC) few-UC CrTe crystals with pre-magnetized AFM tip. **c, d** MFM phase angle distribution (**d**) of selected regions for substrate (red), single tri-UC CrTe (few-UC-1, green) and two stacked tri-UC CrTe (few-UC 2, blue) for in (**c**) with 300, 100, and 50 nm side width (up to down in MFM phase image in (**c**)). **e** MFM phase angle difference in substrate, few-UC-1, and few-UC-2 CrTe crystals with side width from 10 to 300 nm. No external magnetic field is applied on few-UC CrTe during all the MFM tests. Error bars s.e.m.; $N = 30$.

Figs. 22 and 23). The few-UC CrTe crystal is stable even after one-month exposure in ambient air as shown in XRD (Supplementary Fig. 22) and the strong intrinsic ferromagnetism in both multi-UC and few-UC CrTe crystals still remained (Supplementary Fig. 23a–d). The freestanding non-van der Waals ultrathin ferromagnetic 2D crystals may provide new and unique platforms for room-temperature even beyond-room-temperature intrinsic 2D ferromagnetism, magnetic junction, integrated 2D systems from large numbers of non-van der Waals crystals, and van der Waals/non-van der Waals 2D heterostructures.

## Discussion
In this work, we first theoretically calculate the cleavage energy of different crystal planes in CrTe and predict the cleavage direction. The small cleavage energy of (002) face allows us to exfoliate CrTe-like van der Waals materials. To get freestanding few-UC and multi-UC CrTe crystals, we develop a CVD-assisted ultrasonication method, which allows us to obtain a large number of 2D CrTe nanosheets in freestanding form for potential applications such as spintronics, magnetic resonance imaging, and ferrofluid. Freestanding ultrathin 2D CrTe crystals with thickness down to mono-UC, few-UC, and multi-UC are acquired by this CVD-assisted ultrasonication method.

The Curie temperature ($T_C$) of freestanding 2D CrTe crystals is above room temperature and higher than bulk CrTe crystal, which is notably different from van der Waals intrinsic ferromagnetic 2D crystals where $T_C$ is lower in few-layer or mono-

layer crystals than in bulk crystals. The results herein open an alternative avenue to achieve intrinsic, strong, room-temperature, or beyond-room-temperature 2D ferromagnetism in freestanding ultrathin 2D crystals.

## Methods
**Preparation of ultrathin CrTe 2D crystals**. Briefly, a ceramic crucible with 0.1–0.6 g $CrCl_3$ powder was put into the center of the tube and a ceramic crucible with 0.1–0.5 g Te powder was placed upstream away from $CrCl_3$. A mica or $SiO_2$/Si substrate was placed downstream from the center of $CrCl_3$. Growth occurred at atmospheric pressure in an argon/hydrogen (5:1 in volume ratio) mixture environment for over 2 h. The furnace temperature for growth was optimized in the range of 650–1100 °C (typically, over 900 °C) as shown in Supplementary Fig. 1, and then the furnace cooled down naturally. The obtained CrTe crystals are shown in Supplementary Fig. 2. To obtain multi-UC CrTe crystals, the crystals on substrate are easily removed from substrate, collected and dispersed in ethanol, followed by shaking for 5 min and sonicating for 5 min. Then the solution was placed stirlessly for 20 min to let the thick piece sink, then the supernatant with multi-UC CrTe crystals was transferred and these multi-UC CrTe crystals were collected by filtration. To get few-UC CrTe crystals, the same solution was sonicated in cold water (5 °C) for 12 h and placed for 24 h. Bulk CrTe crystal is prepared by heating mixed 1 g $CrCl_3$ and 1 g Te powders for 5 h in a quartz tube under the same conditions described above. To understand the formation of ultrathin 2D CrTe crystals with exposed (002) faces, the exfoliation energies ($E_{ex}$) of CrTe nanosheets with four different crystal faces (i.e., (002), (100), (101), and (110) faces) have been calculated as follows (Eq. (1)):

$$E_{ex} = (E_{sheet1} + E_{sheet2} - E_{tot})/A \qquad (1)$$

where $A$ is surface area of a given CrTe nanosheet, and $E_{tot}$, $E_{sheet1}$, $E_{sheet2}$ correspond to the total energies of the thicker CrTe nanosheet and two detached thinner CrTe nanosheets (Fig. 1b–e).

**Characterizations and measurements.** The phase, morphology, components, atomic structures, and thickness of CrTe crystals were characterized by XRD (PANalytical B.V., x'pert3 powder), X-ray fluorescence microprobe (XRF; EDAX Inc., EAGLE III), scanning electron microscope with EDAX attachment (SEM; FEI, Quanta650 FEG), high-resolution electron microscope with EDAX attachment (HRTEM; FEI, Titan G2 60–300), and atomic force microscopy (AFM; Park, XE7), respectively. For MFM imaging, an AFM tip with ferromagnetic coating and a radius of 30 nm was pre-magnetized by NdFeB before imaging. No external magnetic field is applied on CrTe samples during all the MFM tests.

The magnetic properties were measured by a physical property measurement system (PPMS DynaCool, Quantum Design, USA) equipped with a VSM. For VSM sample preparation with good orientations, we drop a CrTe solution onto a PVDF (polyvinylidene fluoride) or carbon millipore filter for sheet orientation control as shown in Fig. 2c and Supplementary Fig. 5a, b. The sheets are easily oriented parallel to a millipore filter substrate due to their 2D sheet structures after natural drying (Fig. 2c and Supplementary Fig. 5a, b). To determine the mass of collected multi-UC and few-UC crystals, we weighed the millipore filter before and after CrTe crystals deposition. To gain highly oriented CrTe crystals, the amount of the deposited CrTe crystals are controlled in the range of 2~3 mg (electronic balance with weight sensitivity, 0.01 mg). Then we fix the millipore substrate parallel to the VSM quartz sample holder by a PTFE (poly tetra fluoroethylene) micro-film winding. We note that magnetic signal from PTFE micro-film is also included in background signal test in Supplementary Fig. 7, so it can be ignored for ferromagnetism tests. In order to get more reliable magnetization results, we prepare three different batches of samples for magnetization measurement. For MT curve, each of the point is set to measure 400 times for average and 200 times for average for MH curve. Except the magnetic anisotropy measurement in Supplementary Fig. 9, all of the magnetization measurements are carried out with magnetic field in-plane oriented because of the in-plane easy magnetization. In some tests, measurements of bulk CrTe crystal were carried out under the same conditions as multi-UC and few-UC CrTe crystals. To test the sample with spontaneous magnetization regime ($B=0$), the samples are firstly heated to 400 K (above the $T_C$) and held for 5 min. Then the magnetic field is raised to 2 T and decreased to 0 T with oscillated mode to remove the remanence of the samples and the superconducting coil. After that, the temperature is reduced to 2 K at a rate of 2 K/min under zero field ($B=0$). During the cooling process, the magnetic moment of the sample is continuously measured by VSM with a vibration frequency of 40 Hz and amplitude of 2 mm.

Single sheet magneto-optical Kerr effect tests for single sheet bulk and multi-UC CrTe crystals under different temperature were done by a magneto-optical Kerr Microscope & Magnetometer (Evico) with a vacuum stage for varied temperature control. Note that single sheet few-UC CrTe crystals is beyond the sensitivity of magneto-optical Kerr microscopy used here and MFM are tested for them.

**Theoretical calculations.** The electronic and magnetic properties of CrTe bulk and ultrathin nanosheets have been investigated by using the first-principles plane-wave method within DFT framework, as implemented in the Vienna ab initio simulation package (VASP)[29,30]. The electron–ion interactions were represented by all-electron projector-augmented wave (PAW) potential[31]. The exchange-correlation energy was treated by generalized-gradient approximation (GGA) of Perdew-Burke-Ernzerhof (PBE)[32]. However, the GGA functional fails to describe partially occupied $d$ and $f$ electronic states of transition-metal elements. Therefore, the GGA with the correction of a Hubbard $U$ term was used in the present study. For the Hubbard parameter $U_{eff} = U - J$[33], we use a typical value of 2 eV that was proved a reasonable value to describe the correction energy of localized Cr-3$d$ states[34]. The kinetic energy cutoff of the plane-wave basis set was considered to be 500 eV. The structures were optimized using the conjugate gradient scheme until the forces on every atom are smaller than $10^{-2}$ eV/Å. The k-point sampling in the Brillouin zone was implemented by the Monkhorst-Pack scheme with the grids from $16 \times 16 \times 16$ to $16 \times 16 \times 1$ for CrTe crystals with different thickness, respectively. To avoid artificial interaction among the neighboring unit cells, a vacuum layer of ~15 Å was included in the model of nanosheets perpendicular to the growth direction. In the calculations, the model of CrTe nanosheets was established by cutting from the optimized CrTe bulk along the (001) direction with desired thickness. The optimized lattice constants of CrTe bulk are $a = b = 4.005$ Å and $c = 6.242$ Å, which agree well with the experimental values ($a = 3.997$ Å and $c = 6.222$ Å) and previous theoretical calculations[35]. To study the disorder effect on the electronic and magnetic properties of ultrathin CrTe nanosheets, a ($5 \times 5$) supercell with a thickness of ~5.8 Å was used to construct the disordered CrTe structure and the k-point sampling in the Brillouin zone was set to $4 \times 4 \times 1$.

## Data availability
The data that support the findings of this study are available from the corresponding author upon reasonable request.

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

## Acknowledgements

This work is funded by National Key Research and Development Program of China (No. 2016YFB0700702), National Basic Research Program of China (No. 2015CB258400), National Natural Science Foundation of China (No. 51402118, 51502101, 61674063), and Zhejiang Provincial Natural Science Foundation of China (No. LD19E010001).

## Author contributions

H.C. designed the project. H.W., J.L., and L.Y. prepared the 2D crystals and did physical properties tests. J.W., L.Z., and H.S. did the theoretical calculations. L.L. and Y.G. carried out the HRTEM. H.C., W.Z., H.W., H.S., and J.D. wrote the paper.

## Competing interests

The authors declare no competing interests.
