## [Peer Review File. · Nature Communications]

Reviewers' Comments:

Reviewer #1:

Remarks to the Author:

This work shows the interesting strategy to fabricate the ferromagnetic 2D materials and the intensive analyses have been performed. However, the current study does not actually provide the magnetic properties of 2D materials. If this work wants to deal with the 2D ferromagnetism, the authors should do more investigations and experiments due to its lack of the 2D ferromagnetic property. Furthermore, 2D crystal should be perfectly manipulated (its thickness and orientation) as well.

1. In Nature Communications, abstract is supposed to be within 150 words and unreferenced. The current abstract is now too long and more seriously it fails to emphasize the interesting issue in the current study.

2. The terminology needs to be reconsidered. "Few-UC" and "Multi-UC" are not scientific at all. Together with this wrong terminology, the authors must define "the exact number of layers" because the quantum phenomena and magnetic properties can be drastically changed even by a single unit-cell thickness.

3. Figure 1g is HRTEM image from few UC sample. If its thickness is assumed to be within 3 layers, the HRTEM image is supposed to provide the phase information of Cr and Te atoms because the weak phase object approximation can be applied. That means there should be the ordered contrast pattern by Te atom with a higher atomic number. On the other hand, figure 1g shows no periodic pattern, which indicates that Cr and Te atoms are not crystallographically, atomically arrayed with a long-range order parameter. If it is not ordered, the origin of the current work will falter. To this end, how about doing HAADF-STEM imaging or (X-ray or neutron) diffraction refinement?

4. How can the authors to normalize the magnetic moments (by emu/g) in cases of Multi-UC and Few-UC samples (Figure 2b~f) by dividing the amount of samples? The magnetic properties should be dealt in much more detail.

5. Because this paper is about ferromagnetic 2D materials, it is necessary to define the direction of magnetic field (i.e., along the in-plane direction or along the out-of-plane direction). There will be bunch of information about the 2D magnetic properties such as the magnetic anisotropy and the subsequent spin Hall effect.

6. The most serious point is that there's no indication of whether the magnetic property is intrinsic or extrinsic. Since a CrTe crystal is not a van der Waals crystal, it can not be simply free of defects at the surface. In this CrTe crystal, the magnetic property can be changed by the type of termination layer (Cr termination, Te termination, half unit-cell termination, and so on..). If the authors want to insist on the intrinsic properties, the surface structure should be defined (for example, by STM).

6. Figures 1~3 are all not clear and thus the author cannot represent the meaning of the results. There are too many sub-figures but there are no link between them in each figure. I ask the authors to choose the most important data which are fitting into the main concept of each figure. FYI, the font sizes are too small to recognize some of them.

7. I don't think figure 4 is quite necessary in the main figure set at the moment. The author may move it into the supporting information. In my opinion, the authors must more concentrate on unveiling (1) whether it is intrinsic or extrinsic and (2) the typical, representative magnetic properties in the field of 2D ferromagnetism.

Reviewer #2:

Remarks to the Author:

The authors report a systematic study on the magnetic properties of CrTe, one of the traditional

ferromagnetic compound They report a results on the few UC CrTe for having a 367 K T_c , together with a comparison with bulk CrTe and other element materials. However, from the point of view of future application and its impact on the intrinsic property of CrTe, I don't think this work deserve a publication on Nature Communications. The reasons listed below:

1. Room-temperature for NiAs type CrTe is a well-known results in its study, several work already were published(as mentioned in the introduction). Even though the author claimed an increase of around 20 K in few UC CrTe, however, the definition of few UC is so rough to get an accurate conclusion. Even though, the magnetic property of CrTe can be enhanced at 1-2 UC limits, the thickness control for this non-Van Der Waals materials to grow a wafer-size towards future interface engineering, similar as epitaxy a UC thick 2D-crystal with wafer scale, is always a big challenging for community. In fact, here, because CrTe is not a typical Van Der Waals layer materials, more difficulty is expected int the future interface engineering.

2. For the magnetic property owned by CrTe itself, from this work, I didn't see any surprising results coming out (except a 20 K enhancement which is of bad quality control). Specially, the CrTe still don't show perpendicular magnetic anisotropy in the 2D limits (its in-plane anisotropy was already been reported else, I think the author should cite the previous work), which is of critical importance for the future spintronic application, for example, resealizing strong magnetic proximity effect into adjacent other 2D materials. Therefore, there is no significant novelty in the intrinsic property of CrTe being revealed by this research.

3. Regarding to the research methods, the author did't pay great attention on the epitaxy, however, they exfoliated the thin from the substrate and break them into small pieces for studying the magnetic property. I want to say, this kind research is not targeted for the application CrTe in future spintronics, as everybody know, to achieve a function device, we need epitaxy of several compounds with sharp interface over the wafer size, only in this situation, we can obtain real applicable device array. Therefore, this research didn't represent a work with future promising on the application of CrTe

Overall, I didn't agree with the publication of this work at current form. I suggest the author pay more attention on the growth of CrTe with fine thickness control on the wafer, and tune the magnetic property via epitaxy engineering, as many efforts have been input into this direction (the author should do more literature survey on this direction).

Response to referees:

Reviewer #1:

This work shows the interesting strategy to fabricate the ferromagnetic 2D materials and the intensive analyses have been performed. However, the current study does not actually provide the magnetic properties of 2D materials. If this work wants to deal with the 2D ferromagnetism, the authors should do more investigations and experiments due to its lack of the 2D ferromagnetic property. Furthermore, 2D crystal should be perfectly manipulated (its thickness and orientation) as well.

Reply: Thanks reviewer for recognizing the importance and novelty of our work.

(1) Actually, there are four major notable points of about 2D CrTe crystal and its ferromagnetism in this work:

(I) Freestanding ultrathin non-van der Waals 2D CrTe crystals with thickness down to mono, few and multi-unit cell (UC) are acquired by a CVD-assisted ultrasonication method, showing great potential to prepare freestanding non-van der Waals intrinsic ferromagnetic 2D crystals, study their 2D quantum confinement effect and be applied in spintronics, magnetic resonance imaging, ferrofluid and drug carrier. Note that freestanding mono-UC non-van der Waals

2D CrTe crystals do exist and is obtained by this reported new method. See more details in notes for point 1 below.

(II) The Curie temperature (T_c) of freestanding 2D CrTe crystals is higher than bulk CrTe crystal, which is unexpectedly contrast with the discovered van der Waals intrinsic ferromagnetic 2D crystals which has much lower T_c than bulk crystal (Ref. 11-13, 15, Nature **546, 265-269(2017); Nature **546**, 270-273(2017); Nat. Mater. **17**, 778-782(2018); Nat. Mater. **17**, 406-410(2018)), due to 2D quantum confinement effect induced enhancement of spin polarization with reduction of thickness and dimension. We also clearly show this unexpected opposite T_c behavior in freestanding non-van der Waals 2D crystals versus van der Waals ferromagnetic 2D crystals at single sheet level, consistent with the VSM results. The results herein open a new unexpected avenue and enable to achieve intrinsic, strong, room temperature or beyond room temperature 2D ferromagnetism in freestanding ultrathin 2D crystals for the first time, without any requirement for extrinsic modulations.**

(III) The strong intrinsic ferromagnetism of freestanding 2D CrTe crystals are robust at room temperature and the Curie temperature T_c is up to 367 K. This is surprisingly contrast with almost all the discovered van der Waals intrinsic ferromagnetic 2D crystals which show intrinsic 2D ferromagnetism only below room temperature. The strong intrinsic room temperature even beyond room

temperature 2D ferromagnetism of freestanding 2D CrTe crystals comparable with many widely-used traditional materials such as Fe, Co, Ni, and BaFe₁₂O₁₉ indicates a great potential in 2D spintronics, magnetic resonance imaging, and ferrofluid. More discussion can be found in **notes for point 3** below.

(IV) Due to the 2D quantum confinement effect and resulted enhancement of spin polarization, reduction of thickness and dimension induces significant changes in other ferromagnetic properties including spontaneous magnetization (B=0) or spontaneous spin polarization, variable saturation magnetization, highly tunable coercivity and room temperature linear domain wall (LDW) magnetic domain behavior in freestanding 2D CrTe crystals. 2D CrTe crystals show up to 3.7 times larger dramatic enhancement in spontaneous magnetization (magnetization without external magnetic field, B =0; highly related with the spontaneous spin polarization of the conduction electrons or carriers) **for the 2D quantum confinement effect induced enhancement of spin polarization with reduction of thickness and dimension, providing a new route to tune and achieve high spin polarization in freestanding 2D ferromagnetic crystals.** 2D CrTe crystals also show up to 35% larger saturation magnetization and 10.5 times larger coercivity than bulk CrTe crystals. In addition, **room temperature linear domain wall (LDW) magnetic domain behavior** important for magnetoelectronics (Supplementary Fig. 10 in Supplementary Materials, or see notes for point 4 below) is observed in 2D CrTe crystals rather than random domain walls observed in bulk CrTe crystals.

Notes for point 1: Freestanding ultrathin non-van der Waals 2D CrTe crystals down to mono-UC and few-UC are obtained by a CVD-assisted ultrasonication method. As shown in **Fig. 1h-j** (see below), the thickness can be 0.8, 2, and 3 nm, corresponding to mono-, bi- and tri-UC, respectively, considering theoretical thickness of mono-UC CrTe 0.624 nm. Mono-UC non-van der Waals 2D CrTe crystals do exist and is obtained by this method. This means rarely-studied freestanding non-van der Waals 2D crystals have great potential in 2D electronics and spintronics like recently intensely studied van der Waals 2D crystals. Freestanding non-van der Waals 2D ferromagnetic CrTe crystals enable studying 2D quantum confinement effect on ferromagnetic properties in freestanding non-van der Waals 2D crystals.

Fig. 1h-j : AFM images for few-UC CrTe crystals with different thickness of 0.8 nm (h), 2 nm (i) and 3 nm (j).

Notes for point 3: Almost all of the freestanding van der Waals 2D ferromagnetic crystal show intrinsic ferromagnetism lower than room temperature, which greatly limited their practical applications. But freestanding non-van der Waals 2D CrTe crystals show higher Curie temperature beyond room temperature even down to

ultrathin few-UC 2D crystal. The intrinsic high Curie temperature up to room temperature make it easier to be applied in practical fields compared with van der Waals ferromagnetic 2D crystals. In addition, many discovered van der Waals 2D ferromagnetic crystals show unavoidable instability in long-term exposure of ambient air atmosphere while non-van der Waals 2D CrTe crystals show unusual crystal and ferromagnetism stability in long term, ambient air conditions.

Notes for point 4: Spontaneous magnetization, in ultrathin few-UC and multi-UC CrTe is 4.7 and 2.3 times that of bulk CrTe crystal, respectively (**Fig. 3d**). Considering the spontaneous magnetization is highly related with the spontaneous spin polarization of the carriers or conduction electrons(Ref. 29-30, *Physica* **24**, 39-51(1958); *Eur. Phys. J. B* **36**, 593-598(2003)), the results indicate the spin polarization of freestanding 2D CrTe crystals increase significantly with decreasing thickness. It is consistent with our theoretical first-principles calculations which imply that the spin polarization is significantly enhanced especially for electronic states around Fermi level with the decrease of thickness and dimension.

As for room temperature linear domain walls(LDW), the boundary of the magnetic domain on freestanding 2D CrTe crystals is always straight line, as shown in newly added magnetoptical Kerr effect tests and microscopy of single sheet bulk and multi-UC CrTe crystals (**Supplementary Fig. 10d-g** , see below). Such linear boundary between magnetic domain is called linear domain walls (LDW) proposed by

Bulaevskii and Ginzburg (Soviet Physics JETP *18*, 530-534(1964)). Random domain wall behavior is always observed in bulk CrTe crystals and no LDW is found in them (Fig. S11).

Supplementary Fig. 10: Single sheet magnetooptical Kerr effect tests and microscopy of single sheet bulk and multi-UC CrTe crystals, showing higher T_c in multi-UC than bulk CrTe crystals (A-C) and room temperature LDW magnetic domain wall behavior in multi-UC CrTe 2D crystal. (A-B) Kerr intensity curves for a single sheet bulk (A, 1.1 μm , ~ 1375 UC) and multi-UC (B, 57 nm, ~ 71 UC) CrTe crystals at different temperatures measured by Evico magnetics Kerr Microscope & Magnetometer. (C) Single sheet magnetooptical Kerr effect tests-derived T_c of bulk and multi-UC CrTe crystals by single sheet tests. Error bars sem; $N=3$. (D-E) Room temperature magnetooptical Kerr images for two multi-UC CrTe single sheets at magnetic field from 0 mT to 10 mT, the dash lines shows the linear domain walls. Thickness of sheets in (D) and (E) is 57 nm (~ 71 UC) and 36 nm (~ 45 UC), respectively. (F-G) Room temperature Kerr intensity curves collected from multi-UC CrTe single sheets in (D) and (E).

(2) As for the thickness control, we can achieve certain content of control for the manipulation. Through our CVD-assisted ultrasonication method, both single, double

and 3-unit cell thick CrTe crystals can also be obtained easily. However, like mechanically-exfoliation method by scotch tape used in many reports, we can achieve the control in some certain sheets, but in large or mass scale, it is difficult to achieve large scale sheets in exact one thickness, like mechanically-exfoliated sheets by Scotch tape where one sheet usually contains several parts with different thickness. Whereas we can still use CVD-assisted ultrasonication method to obtain CrTe nanosheets with very narrow thickness distribution (mostly distributed in 2-3 nm, that is bi- or tri-UC). Our main goal is to study the properties of 2D ferromagnetic nanosheets in free-standing form which may be potential for applications such as magnetic resonance imaging and ferrofluids where a lot of freestanding nanosheets are required.

As for the orientation manipulation, we slowly drop CrTe nanosheet crystals dispersed in ethanol solvent onto the SiO₂/Si or milliporous substrate. The 2D CrTe crystals are easily oriented parallel to the substrate for their 2D ultrathin nature. The plane of the CrTe nanosheet on the substrate is parallel to the plane of the substrate (Fig. 2c, Supplementary Fig. 5a, b), which allows us to manipulate the orientation of the CrTe nanosheet crystals by manipulating the substrate. As shown in manuscript and supporting information, we also test the ferromagnetism of highly oriented single sheets by magnetic force microscopy (MFM, Fig. 4) and magnetooptical Kerr effects (Supplementary Fig. 8 and Fig. 10).

Fig. 2a: Free-standing few-UC CrTe crystals are well dispersed in ethanol, attracted by a ferromagnet and well re-dispersed after attraction.

Fig. 1h-j: AFM images for few-UC CrTe crystals on SiO₂/Si substrate with different thickness of 0.8 nm (h), 2 nm (i) and 3 nm (j).

Figure Rearranged from Fig. 2c and Supplementary Fig. 5: Few-UC CrTe crystal on milliporous substrates. **a, b, d,** Micrometer-Scale TEM pictures of the few-UC CrTe crystals. **e,** The corresponding selected-area electron diffraction (SAED) of (a).

1. In Nature Communications, abstract is supposed to be within 150 words and

unreferenced. The current abstract is now too long and more seriously it fails to emphasize the interesting issue in the current study.

Reply: Thanks for reviewer's comments. We have revised the abstract to the following:

“Ferromagnetism control is one of the most important fundamentals for modern electronics and spintronic, topological and quantum technologies in future. Inducing long-range ferromagnetic order in ultrathin 2D crystals will provide more functional possibility to combine their unique electronic, optical and mechanical properties to develop new multifunctional coupled applications. Recently discovered intrinsic 2D ferromagnetic crystals such as $\text{Cr}_2\text{Ge}_2\text{Te}_6$, CrI_3 and Fe_3GeTe_2 are intrinsically ferromagnetic only below room temperature, mostly far below room temperature (T_c , ~20-207 K). Here we develop a scalable method to prepare freestanding non-van der Waals ultrathin 2D crystals down to mono- and few unit cells (UC) and report unexpected strong, intrinsic, ambient-air-robust, room-temperature ferromagnetism with T_c up to ~367 K in freestanding non-van der Waals 2D CrTe crystals. Freestanding 2D CrTe crystals show comparable or better ferromagnetic properties to widely-used Fe, Co, Ni and $\text{BaFe}_{12}\text{O}_{19}$, promising as new platforms for room-temperature intrinsically-ferromagnetic 2D crystals and integrated 2D devices.”

2. The terminology needs to be reconsidered. "Few-UC" and "Multi-UC" are not scientific at all. Together with this wrong terminology, the authors must define "the exact number of layers" because the quantum phenomena and magnetic properties can be drastically changed even by a single unit-cell thickness.

Reply: For a van der Waals crystal, since it is a layered structure, people are used to using "layer" to describe its thickness. But for non-van der Waals crystals, they do not have a layered structure, so "unit cell (UC)" is often used to evaluate their thickness (Nat. Mater. **9**, 397–402 (2010); Nat. Mater. **14**, 801–806 (2015); Phys. Rev. Lett. **103**, 146101 (2009); Phys. Rev. Lett. **102**, 176805 (2009); Phys. Rev. Lett. **119**, 256404 (2017); Sci. Rep. **4**, 6040 (2014)).

In freestanding form, the ultrasonically-splitted CrTe nanosheets must contain a certain thickness range. As shown below, the few-UC is defined as the thickness between 0.8 – 10 nm (1 - 12 UC), and mainly distributed in 2 - 4 nm (2 – 5 UC). The multi-UC is defined as the thickness between 10 - 60 nm (12 – 75 UC) and mainly distributed in 20 - 40 nm (25 – 50 UC). We have annotated their thickness in the discussions.

We note that, contrast with van der Waals 2D ferromagnetic crystals, the sharp changes with thickness are not observed in the non-van der Waals CrTe 2D crystals.

Figure Rearranged from Fig. 2b and Supplementary Fig. 2b: Thickness distribution of few-UC (left) and multi-UC (right) CrTe crystals from AFM statistical analysis.

3. Figure 1g is HRTEM image from few UC sample. If its thickness is assumed to be within 3 layers, the HRTEM image is supposed to provide the phase information of Cr and Te atoms because the weak phase object approximation can be applied. That means there should be the ordered contrast pattern by Te atom with a higher atomic number. On the other hand, figure 1g shows no periodic pattern, which indicates that Cr and Te atoms are not crystallographically, atomically arrayed with a long-range order parameter. If it is not ordered, the origin of the current work will falter. To this end, how about doing HAADF-STEM imaging or (X-ray or neutron) diffraction refinement?

Reply: As shown in Fig 1e-f or below, the selected area electron diffraction (SEAD) of CrTe nanosheet crystal is perfect hexagonal lattice arrangement, which indicates a well crystalline and uniformity of thickness of the CrTe nanosheet crystal. The reduced FFT image from the SEAD also indicates the high crystallinity of CrTe nanosheet 2D crystal. Our HRTEM also shows clear lattice fringes of (110) plane. But due to strong room temperature ferromagnetism influence of CrTe nanosheet 2D

crystal on the electron focusing, we couldn't clearly see the atomic distribution to distinguish Cr and Te atoms.

Fig. 1e-f: **e**, HRTEM image of the few-UC CrTe crystal; **f**, The selected-area electron diffraction (SAED) for the few-UC CrTe crystal. **g**, Reduced Fast Fourier Transformation (RFFT) image derived from (**f**).

4. How can the authors to normalize the magnetic moments (by emu/g) in cases of Multi-UC and Few-UC samples (Figure 2b~f) by dividing the amount of samples? The magnetic properties should be dealt in much more detail.

Reply: We've showed the details about normalizing the magnetic moments in our Supplementary Information. As mentioned in Characterizations and measurements in Supplementary Information, we drop a mass of 2D CrTe crystal solution onto the millipore filter substrate as Fig. 2c and Supplementary Fig. 5 show. The 2D CrTe crystals are easily oriented parallel to the substrate for their 2D ultrathin nature. The millipore filter substrate has been measured to be diamagnetic to preclude any influence and thus has no contribution to the ferromagnetism of the samples (Supplementary Fig. 6). To determine the mass of collected multi- and few-UC crystals, we weigh the millipore filter substrate before and after CrTe 2D crystals

deposition with an electronic balance of high precision (0.01mg). To gain highly oriented CrTe crystals, the amount of the deposited 2D CrTe crystals are controlled in range of 2~3 mg. In order to get more accurate magnetization results, we prepare three different batches of samples for magnetization measurements.

5. Because this paper is about ferromagnetic 2D materials, it is necessary to define the direction of magnetic field (i.e., along the in-plane direction or along the out-of-plane direction). There will be bunch of information about the 2D magnetic properties such as the magnetic anisotropy and the subsequent spin Hall effect.

Reply: As mentioned above, we have shown the anisotropy of our few-UC and multi-UC CrTe crystals in Supplementary Fig. 8a, b, both of which show in-plane preferred easy-axis. Furthermore, we have also verified the in-plane easy-axis of a single sheet multi-UC CrTe crystal through magnetooptical Kerr rotation tests, as shown in Supplementary Fig. 8c, d below. Due to the in-plane preferred easy-axis of our CrTe crystals, all of other magnetic measurement were carried out with a-b plane parallel to the magnetic field (in-plane alignment). Due to the in-plane magnetization feature, anomalous Hall effect cannot be observed due to the Hall effect requires sample plane perpendicular to the magnetic field. However, although no anomalous Hall signal is detected by electrical method, we have detected the Kerr rotation signals by the magnetooptical method. As shown in Supplementary Fig. 8, the CrTe crystals show clear ferromagnetic Kerr rotation behavior with in-plane easy-axis.

Supplementary Fig. 8: Magnetic anisotropy of multi- and few-UC CrTe crystals. **a, b,** The in-plane and out-of-plane M-H hysteresis loops for multi-UC (**a**) and few-UC (**b**) CrTe crystals at 300 K. **c, d,** The out-of-plane (**c**) and in-plane (**d**) Magneto-optical Kerr rotation loops for a single sheet multi-UC CrTe crystal at 300 K.

6. The most serious point is that there's no indication of whether the magnetic property is intrinsic or extrinsic. Since a CrTe crystal is not a van der Waals crystal, it can not be simply free of defects at the surface. In this CrTe crystal, the magnetic property can be changed by the type of termination layer (Cr termination, Te termination, half unit-cell termination, and so on..). If the authors want to insist on the intrinsic properties, the surface structure should be defined (for example, by STM).

Reply: (1) Regarding the question of whether it is intrinsically ferromagnetic, we would like to address this from the following 4 points. **First**, the bulk CrTe crystal is an intrinsic ferromagnetic material as reported and confirmed by VSM tests in Fig. 3a and Fig. 3d. **Second**, spontaneous magnetization regime tests without external

magnetic field ($B=0$, zero magnetic field) of 2D CrTe crystals show clear transition at the Curie temperature, similar with FC-ZFC magnetization measurements, and further confirm its intrinsic feature (Fig. 3d). **Third**, The saturation magnetic moment of CrTe nanosheets obtained in our work is close to 100 emu/g, which is $10^5\sim 10^9$ times that of defect-induced magnetism (generally $\mu\text{emu/g}$ to memu/g , Appl. Phys. Lett. **104**, 202406(2014)), implying that our multi- and few-UC crystals are more like intrinsic ferromagnetism, not defects-induced ferromagnetism. **Finally**, the CrTe crystals have a magnetooptical Kerr rotation effects on polarized light, which also implies its intrinsic ferromagnetic properties from optical perspective.

Fig. 3d Spontaneous magnetization regime tests without external magnetic field ($B=0$) for bulk, multi-UC and few-UC CrTe crystals from 2 K to 400 K.

(2) As for the effect of terminated atoms on ferromagnetism of CrTe crystals, we have revealed that the ferromagnetic moments of ultrathin 2D CrTe crystals mainly come from 3d orbital of Cr atom, as shown in Supplementary Fig. 13. Te atoms have negligible contribution to the ferromagnetism. From Supplementary Fig. 16, we can see that the magnetic moment of outer Cr atoms are slightly larger than inner Cr

atoms. If the exposed surfaces are terminated by Cr atoms, we will expect slightly larger magnetic moment. Otherwise, for Te termination, the magnetic moment will slightly smaller than Cr- terminated one. Moreover, the VSM results show the saturated magnetic moment of few-UC CrTe 2D crystals up to 89 emu/g , $10^5 \sim 10^9$ times larger than typical defect induced ferromagnetic moment, which cannot be dominated by termination-induced defect ferromagnetism.

Supplementary Fig. 13: The spin-resolved density of states (DOSs) and spin density distributions of bulk, multi-UC, few-UC and disordered few-UC CrTe crystals. a-d, Total DOS and partial DOS of Cr-3d states in bulk (a), multi-UC (b), few-UC (c) and disordered few-UC (d) CrTe crystals. The vertical dash lines denote the position of Fermi level. **e, f,** Spin density distribution of ordered few-UC (e) and disordered few-UC (f) CrTe crystal. The red and blue isosurfaces represent positive and negative spin density, respectively.

Supplementary Fig. 16: The calculated atomic magnetic moments of bulk (a), multi-UC (b) and few-UC (c) CrTe crystals.

(3) For the exposed surface, since the multi/few-UC crystals are a collection of ultrasonication-exfoliated nanosheets, which are obtained by splitting thick crystals along a specific crystal plane of lowest exfoliation energy, then there must be both Cr atoms exposed surfaces in some sheets and Te atoms exposed surfaces in others. For the CrTe single crystals directly grown by CVD, the surface energy of the (002) plane by Te termination is the lowest according to our theoretical calculations as discussed below with more detailed analysis. We also note that it is very hard to get perfectly deposited samples from freestanding ultrathin sheets in solution for STM tests to identify the atoms of top layer surface.

To simulate the surface of NiAs-type CrTe, we adopt a slab model with 17 atomic layers and a vacuum region of 12Å to avoid interaction with each other (Fig. R1a,b). The (002) surface can be terminated either by Cr atomic layer or Te atomic layer (labeled as (002)-Cr and (002)-Te respectively). First, the topmost five layers are relaxed. Based on the relaxed structure, we calculated the surface energy to study the relative stability of the surface. The surface energy can be expressed as:

$$E^{surf} = \frac{1}{A} [E_{tot}^{slab} - N_{Te} E_{CrTe}^{bulk} + (N_{Te} - N_{Cr}) \mu_{Cr}],$$

Where A is the surface area, E_{tot}^{slab} is the total energy of the slab, E_{CrTe}^{bulk} is the energy of bulk CrTe, N_{Cr} and N_{Te} are numbers of Cr and Te atoms in the slab, and μ_{Cr} is

the chemical potential of Cr. μ_{Cr} is related with the chemical potential of Te μ_{Te} with :

$$\mu_{Cr} + \mu_{Te} = \mu_{CrTe}^{bulk},$$

The Cr and Te chemical potential in the slab must be less than the corresponding elemental bulk chemical potential:

$$\mu_{Cr} \leq \mu_{Cr}^{bulk}, \mu_{Te} \leq \mu_{Te}^{bulk}.$$

The variation of μ_{Cr} is:

$$\mu_{CrTe}^{bulk} - \mu_{Te}^{bulk} \leq \mu_{Cr} \leq \mu_{Cr}^{bulk}.$$

The result shows that the Te-terminated (002) surface has lower surface energy than Cr-terminated (002) surface over the effective chemical potential range, hence the Te-terminated surface are energetically more stable (Fig. R1c).

Fig. R1: The slab model for the surface structure of NiAs-type CrTe and corresponding surface energies. a, The Cr-terminated (002) surface. b, The Te-terminated (002) surface. c, Surface energies as a function of chemical potential of Cr for the Cr and Te terminated (002) surface.

7. Figures 1~3 are all not clear and thus the author cannot represent the meaning of the results. There are too many sub-figures but there are no link between them in each figure. I ask the authors to choose the most important data which are fitting into the main concept of each figure. FYI, the font sizes are too small to recognize some of them.

Reply: Some displayed figures may be too small to be effectively recognized as reviewer said. We have put unimportant figures in the supplementary information and enlarged the remained figures. As for Fig. 1, we mainly intend to show the crystal structure of non-van der Waals CrTe, indicating that the (002) plane is easy to break, and show the ultrathin 2D CrTe crystal obtained by CVD-assisted ultrasonication method. Fig.2 shows characterizations for few-UC CrTe crystals. Fig. 3 shows the ferromagnetism of three kinds of thicknesses of CrTe crystals and their comparison with traditional ferromagnetic materials. Fig. 4 shows room temperature ferromagnetism of few-UC CrTe crystals by single sheet MFM tests at nanoscale.

8. I don't think figure 4 is quite necessary in the main figure set at the moment. The author may move it into the supporting information. In my opinion, the authors must more concentrate on unveiling (1) whether it is intrinsic or extrinsic and (2) the typical, representative magnetic properties in the field of 2D ferromagnetism.

Reply: Figure 4 in original manuscript mainly talks about the relatively stable

ferromagnetic moment of the multi- and few-UC CrTe crystals under the ambient environment, contrast with the instability of most van der Waals 2D ferromagnetic crystals in air. As suggested, we have remove this part to the supplementary information. Regarding whether the magnetic moment of CrTe is intrinsic or extrinsic, all evidence indicate an intrinsic ferromagnetism as discussed in Reply to Question 6. Its magnitude of saturated magnetic moment, the spontaneous magnetization without external magnetic field ($B=0$), and magneto-optical Kerr loops all indicate that it is intrinsically ferromagnetic rather than extrinsically.

Reviewer #2:

The authors report a systematic study on the magnetic properties of CrTe, one of the traditional ferromagnetic compound They report a results on the few UC CrTe for having a 367 K T_c , together with a comparison with bulk CrTe and other element materials. However, from the point of view of future application and its impact on the intrinsic property of CrTe, I don't think this work deserve a publication on Nature Communications. The reasons listed below:

Reply: Thanks reviewer for recognizing the importance and novelty of our work.

Actually, there are four major notable points of our work about this non-van der

Waals 2D CrTe crystals.

(1) Freestanding ultrathin non-van der Waals 2D CrTe crystals with thickness down to mono, few and multi-unit cell (UC) are acquired by a CVD-assisted ultrasonication method, showing great potential to prepare freestanding non-van der Waals intrinsic ferromagnetic 2D crystals, study their 2D quantum confinement effect and be applied in spintronics, magnetic resonance imaging, ferrofluid and drug carrier. Note that freestanding mono-UC non-van der Waals 2D CrTe crystals do exist and is obtained by this reported new method. See more details in notes for point 1 below.

(2) The Curie temperature (T_c) of freestanding 2D CrTe crystals is higher than bulk CrTe crystal, which is unexpectedly contrast with the discovered van der Waals intrinsic ferromagnetic 2D crystals which has much lower T_c than bulk crystal (Ref. 11-13, 15, Nature **546, 265-269(2017); Nature **546**, 270-273(2017); Nat. Mater. **17**, 778-782(2018); Nat. Mater. **17**, 406-410(2018)), due to 2D quantum confinement effect induced enhancement of spin polarization with reduction of thickness and dimension. We also clearly show this unexpected opposite T_c behavior in freestanding non-van der Waals 2D crystals versus van der Waals ferromagnetic 2D crystals at single sheet level, consistent with the VSM results. The results herein open a new unexpected avenue and enable to achieve intrinsic, strong, room temperature or beyond room temperature 2D ferromagnetism in**

freestanding ultrathin 2D crystals for the first time, without any requirement for extrinsic modulations.

(3) The strong intrinsic ferromagnetism of freestanding 2D CrTe crystals are robust at room temperature and the Curie temperature T_c is up to 367 K. This is surprisingly contrast with almost all the discovered van der Waals intrinsic ferromagnetic 2D crystals which show intrinsic 2D ferromagnetism only below room temperature. The strong intrinsic room temperature even beyond room temperature 2D ferromagnetism of freestanding 2D CrTe crystals comparable with many widely-used traditional materials such as Fe, Co, Ni, and $\text{BaFe}_{12}\text{O}_{19}$ indicates a great potential in 2D spintronics, magnetic resonance imaging, and ferrofluid. More discussion can be found in **notes for point 3** below.

(4) Due to the 2D quantum confinement effect and resulted enhancement of spin polarization, reduction of thickness and dimension induces significant changes in other ferromagnetic properties including spontaneous magnetization ($B=0$) or spontaneous spin polarization, variable saturation magnetization, highly tunable coercivity and room temperature linear domain wall (LDW) magnetic domain behavior in freestanding 2D CrTe crystals. 2D CrTe crystals show up to 3.7 times larger dramatic enhancement in spontaneous magnetization (magnetization without external magnetic field, $B=0$; highly related with the spontaneous spin polarization of the conduction electrons or carriers) **for the 2D quantum confinement effect**

induced enhancement of spin polarization with reduction of thickness and dimension, providing a new route to tune and achieve high spin polarization in freestanding 2D ferromagnetic crystals. 2D CrTe crystals also show up to 35% larger saturation magnetization and 10.5 times larger coercivity than bulk CrTe crystals. In addition, **room temperature linear domain wall (LDW) magnetic domain behavior** important for magnetoelectronics (Supplementary **Fig. 10**, or see **notes for point 4** below) is observed in 2D CrTe crystals rather than random domain walls observed in bulk CrTe crystals.

Notes for point 1: Freestanding ultrathin non-van der Waals 2D CrTe crystals down to mono-UC and few-UC are obtained by a CVD-assisted ultrasonication method. As shown in **Fig. 1h-j** (see below), **the thickness can be 0.8, 2, and 3 nm, corresponding to mono-, bi- and tri-UC, respectively, considering theoretical thickness of mono-UC CrTe 0.624 nm. Mono-UC non-van der Waals 2D CrTe crystals do exist and is obtained by this method.** This means rarely-studied freestanding non-van der Waals 2D crystals have great potential in 2D electronics and spintronics like recently intensely studied van der Waals 2D crystals. Freestanding non-van der Waals 2D ferromagnetic CrTe crystals enable studying 2D quantum confinement effect on ferromagnetic properties in freestanding non-van der Waals 2D crystals.

Fig. 1h-j: AFM images for few-UC CrTe crystals with different thickness of 0.8 nm (h), 2 nm (i) and 3 nm (j).

Notes for point 3: Almost all of the freestanding van der Waals 2D ferromagnetic crystal show intrinsic ferromagnetism lower than room temperature, which greatly limited their practical applications. But freestanding non-van der Waals 2D CrTe crystals show higher Curie temperature beyond room temperature even down to ultrathin few-UC 2D crystal. The intrinsic high Curie temperature up to room temperature make it easier to be applied in practical fields compared with van der Waals ferromagnetic 2D crystals. In addition, many discovered van der Waals 2D ferromagnetic crystals show unavoidable instability in long-term exposure of ambient air atmosphere while non-van der Waals 2D CrTe crystals show unusual crystal and ferromagnetism stability in long term, ambient air conditions.

Notes for point 4: Spontaneous magnetization, in ultrathin few-UC and multi-UC CrTe is 4.7 and 2.3 times that of bulk CrTe crystal, respectively (**Fig. 3d**). Considering the spontaneous magnetization is highly related with the spontaneous spin polarization of the carriers or conduction electrons(Ref. 29-30, *Physica* **24**, 39-51(1958); *Eur. Phys. J. B* **36**, 593-598(2003)), the results indicate the spin

polarization of freestanding 2D CrTe crystals increase significantly with decreasing thickness. It is consistent with our theoretical first-principles calculations which imply that the spin polarization is significantly enhanced especially for electronic states around Fermi level with the decrease of thickness and dimension.

As for room temperature linear domain walls(LDW), the boundary of the magnetic domain on freestanding 2D CrTe crystals is always straight line, as shown in magnetoptical Kerr effect tests and microscopy of single sheet bulk and multi-UC CrTe crystals (**Supplementary Fig. 10d-g, see below**). Such linear boundary between magnetic domain is called linear domain walls (LDW) proposed by Bulaevskii and Ginzburg (Soviet Physics JETP *18*, 530-534(1964)). Random domain wall behavior is always observed in bulk CrTe crystals and no LDW is found in them (**Supplementary Fig. 11**).

Supplementary Fig. 10: Single sheet magnetooptical Kerr effect tests and microscopy of single sheet bulk and multi-UC CrTe crystals, showing higher T_c in multi-UC than bulk CrTe crystals (A-C) and room temperature LDW magnetic domain wall behavior in multi-UC CrTe 2D crystal. (A-B) Kerr intensity curves for a single sheet bulk (A, 1.1 μm , ~ 1375 UC) and multi-UC (B, 57 nm, ~ 71 UC) CrTe crystals at different temperatures measured by Evico magnetics Kerr Microscope & Magnetometer. (C) Single sheet magnetooptical Kerr effect tests-derived T_c of bulk and multi-UC CrTe crystals by single sheet tests. Error bars sem; $N=3$. (D-E) Room temperature magnetooptical Kerr images for two multi-UC CrTe single sheets at magnetic field from 0 mT to 10 mT, the dash lines shows the linear domain walls. Thickness of sheets in (D) and (E) is 57 nm (~ 71 UC) and 36 nm (~ 45 UC), respectively. (F-G) Room temperature Kerr intensity curves collected from multi-UC CrTe single sheets in (D) and (E).

1. Room-temperature for NiAs type CrTe is a well-known result in its study, several work already were published (as mentioned in the introduction). Even though the author claimed an increase of around 20 K in few UC CrTe, however, the definition of few UC is so rough to get an accurate conclusion. Even though, the magnetic property of CrTe can be enhanced at 1-2 UC limits, the thickness control for this non-Van Der

Van Der Waals materials to grow a wafer-size towards future interface engineering, similar as epitaxy a UC thick 2D-crystal with wafer scale, is always a big challenging for community. In fact, here, because CrTe is not a typical Van Der Waals layer materials, more difficulty is expected in the future interface engineering.

Reply: Thanks for the advice.

(1) As the reviewer mentioned, CrTe is a traditional ferromagnetic material. However, its easy cleavage (002) plane and the resulting freestanding two-dimensional 2D ferromagnetism have not been reported, and, importantly, as we present in manuscript and discussed above, there are many unique features for freestanding 2D CrTe crystals especially when compared with van der Waals 2D crystals.

First, the increased T_c in CrTe 2D crystals compared bulk CrTe is so contrast from van der Waals 2D crystals which always show significantly decreased T_c with thickness reduction. Therefore this work provides a new way to room temperature 2D ferromagnetic crystals. We note that this is the first, intrinsic room temperature 2D ferromagnetic crystal till this submission, compared with low temperature usually far below room temperature (mostly $T_c \sim 20-207$ K) 2D ferromagnetism in van der Waals 2D ferromagnetic crystals.

Second, the ferromagnetism in 2D CrTe crystals are ambient air-stable, which is also

contrast with poor ferromagnetism stability of most discovered van der Waals 2D crystals.

Third, due to the 2D quantum confinement effect and resulted enhancement of spin polarization, reduction of thickness and dimension induces significant changes in other ferromagnetic properties including enhanced spontaneous magnetization ($B=0$) or spontaneous spin polarization, variable saturation magnetization, highly tunable coercivity and room temperature linear domain wall (LDW) magnetic domain behavior in freestanding 2D CrTe crystals.

(2) As for the interface engineering for non-van der Waals 2D crystals, it is more difficult than van der Waals 2D crystals. However, it does not mean it is not possible and some work already show non-van der Waals-van der Waals integration is quite effective as interfaces between van der Waals materials (Nature **557**, 696–700(2018); Nature **567**, 323–333(2019)).

We also note the interface engineering mentioned by the reviewer is a problem mostly encountered in the field of microelectronics, but the application of our material is not limited to this field. Many fields such as magnetic resonance imaging and ferrofluid need this kind of freestanding ferromagnetic nanomaterials (Fig.2a in revision).

Although the thickness control may be a bigger challenge for 2D CrTe crystals than

than van der Waals materials, as we show later in reply to question 3 of the reviewer (see details in reply to question 3 below), it is still possible to get non van der Waals 2D CrTe thin films in future by CVD and other methods as already demonstrated in other non van der Waals 2D materials (Nat. Rev. Phys. **2**, 347–364(2020); Nat. Mater. **15**, 304–310(2016)). In this work, the few-UC CrTe crystals are obtained by the CVD-assisted ultrasonication method, the thickness of few-UC CrTe crystals has a very narrow thickness distribution (mostly distributed in 2-3 nm, that is bi or tri-UC, marked as few-UC in our manuscript). We show these freestanding CrTe 2D crystals is still ferromagnetic, and some magnetic properties are relatively enhanced, quite contrast with van der Waals intrinsic 2D ferromagnetic crystals.

2. For the magnetic property owned by CrTe itself, from this work, I didn't see any surprising results coming out (except a 20 K enhancement which is of bad quality control). Specially, the CrTe still don't show perpendicular magnetic anisotropy in the 2D limits (its in-plane anisotropy was already been reported else, I think the author should cite the previous work), which is of critical importance for the future spintronic application, for example, resealizing strong magnetic proximity effect into adjacent other 2D materials. Therefore, there is no significant novelty in the intrinsic property of CrTe being revealed by this research.

Reply: (1) The enhancement of Curie temperature T_c in freestanding 2D CrTe crystals is very important when we consider all the intrinsic, van der Waals ferromagnetic 2D

crystals with T_c significantly decreased T_c as thickness reduction, and this work provides a new way to realize room temperature and above room temperature 2D ferromagnetic crystals. We note that this is the first, intrinsic room temperature 2D ferromagnetic crystal till this submission with record T_c up to 367 K, contrast with low temperature usually far below room temperature 2D ferromagnetism (mostly $T_c \sim 20-207$ K) in van der Waals 2D ferromagnetic crystals.

Except enhanced T_c compared with bulk, there are also other important unique features in freestanding 2D CrTe crystals. Spontaneous magnetization and spontaneous spin polarization enhancement, LDW, and tunable H_c for freestanding 2D CrTe crystals are quite different from bulk CrTe, as discussed with more details above and in main manuscript.

(2) In Sreenivasan's previous work (*Thin Solid Films* **505**, 133–136(2006)), 100 nm epitaxial NiAs-type CrTe film on (100) GaAs substrate with ZnTe buffer layers shows in-plane anisotropy with Curie temperature of 330 K. We have quoted this work in revised manuscript (see Ref. 21). Although no perpendicular magnetic anisotropy revealed in our work either, we obtained freestanding non-van der Waals CrTe nanosheet 2D crystals whose thickness can be reduced to one unit cell thick, remarkable for non-van der Waals freestanding 2D crystals. For 2D CrTe crystals whose thickness is mainly distributed in 2-3 nm, its T_c is higher than that of bulk and is above room temperature. This, combined with its air stability, lays the foundation

for large-scale applications where precise thickness control is not a key issue. As mentioned above, this large-scale free standing 2D ferromagnetic nanosheet crystals may not have advantages in spintronic devices, as the reviewer said, but it has considerable advantages in magnetic resonance imaging, ferrofluid and other fields. Moreover, as discussed with more details in the Reply to the question 3 of the reviewer below, it is still possible to improve and modify this CVD-based method to wafer scale thin film on SiO₂/Si substrate in future which is compatible with modern Si electronics.

3. Regarding to the research methods, the author didn't pay great attention on the epitaxy, however, they exfoliated the thin from the substrate and break them into small pieces for studying the magnetic property. I want to say, this kind research is not targeted for the application CrTe in future spintronics, as everybody know, to achieve a function device, we need epitaxy of several compounds with sharp interface over the wafer size, only in this situation, we can obtain real applicable device array. Therefore, this research didn't represent a work with future promising on the application of CrTe

Reply: The work on epitaxial growth of CrTe has also been done by Zhao, Dapeng, et al. as we quoted in original manuscript, but the Curie temperature T_c of the epitaxial CrTe film is only ~200 K (Nano Research *II*, 3116-3121 (2018), Ref. 25), which is much lower than room temperature. We note the epitaxial CrTe thin film on substrate shows an in-plane lattice constant of 3.95 Å, while the freestanding CrTe 2D crystal in

our work show an in-plane lattice constant of 4.03 Å. There is a ~2% difference which may come from the difference of freestanding states. The exposure surface of the epitaxially CrTe film is the (111) face, which is different from our CrTe grown by the steady-state CVD method. Our CVD-based CrTe crystals grow along the (002) plane direction and has room temperature ferromagnetism with T_c larger than 350 K and up to 367 K.

By controlling the CVD process parameters, we can also get continuous 2D CrTe films with modified CVD methods, as shown in the Fig. R2 below. But this is not the main research object of this work: freestanding two-dimensional room temperature ferromagnetic 2D crystals. Freestanding 2D CrTe crystals show quite different properties and more flexible in some integration devices. We also note freestanding 2D crystals may show quite different properties comparing with non-freestanding ones, as also discovered recently in freestanding oxide 2D crystals (Nature **570**, 87–90(2019)). We may focus on wafer-scale CrTe 2D thin film by modified CVD method in next work.

Fig. R2: Optical image of 2D CrTe with morphology transition from single crystals to continuous thin film.

Overall, I didn't agree with the publication of this work at current form. I suggest the

author pay more attention on the growth of CrTe with fine thickness control on the wafer, and tune the magnetic property via epitaxy engineering, as many efforts have been input into this direction (the author should do more literature survey on this direction).

Reply: Thanks for the advice, and we will do more efforts in the mentioned direction in future.

We would like to note again the unique features and importance of freestanding room temperature intrinsically-ferromagnetic CrTe 2D crystals.

First, room temperature and above room temperature 2D ferromagnetism in CrTe 2D crystals is contrast with the ferromagnetism low temperature even far below room temperature in almost all intrinsic ferromagnetic van der Waals 2D crystals (mostly $T_c \sim 20-207$ K). We also note 2D ferromagnetism in non-van der Waals CrTe crystals are very stable which is also contrast with poor ambient stability of most van der Waals 2D ferromagnetic crystals.

Second, we discover that the Curie temperature T_c increases with the decrease of thickness in CrTe crystals, which is opposite from that of van der Waals ferromagnetic crystals. Except enhanced T_c compared with bulk CrTe crystals, there are also other important unique features in freestanding 2D CrTe crystals. Spontaneous

magnetization and spontaneous spin polarization enhancement, LDW, and tunable H_c for freestanding 2D CrTe crystals are quite different from bulk CrTe, as discussed with more details above and in main manuscript. The work here therefore imply that although main efforts of 2D ferromagnetic crystals focus on van der Waals crystals, some attention should be paid to the two-dimensional acquisition of non van der Waals crystals when looking for 2D crystals with high Curie temperature.

Thirdly, our CVD-assisted ultrasonication method has a high reference value for other non-van der Waals 2D crystals and 2D ferromagnetism. Most of known materials have non-van der Waals structure, and we provide an effective solution for the scalable preparation of non-van der Waals 2D crystals. We also show the potential of the modified CVD-based method in wafer scale non-van der Waals 2D thin film in future as demonstrated in reply to question 3.

Finally, when the thickness is reduced to few-UC, room temperature ferromagnetism of 2D CrTe crystals is still robust, and is comparable to the traditional ferromagnetic materials such as Fe, Co, Ni and $BaFe_{12}O_{19}$, showing its potential in magnetic resonance imaging, ferrofluid and other fields.

Reviewers' Comments:

Reviewer #1:

Remarks to the Author:

Thanks for the great efforts to enhance the paper.

However, the crystallographic or atomic-structural understanding is still ambiguous. And there still remain the thickness issues as well. Please see the attached file.

Response to referees:

Reviewer #1:

This work shows the interesting strategy to fabricate the ferromagnetic 2D materials and the intensive analyses have been performed. However, the current study does not actually provide the magnetic properties of 2D materials. If this work wants to deal with the 2D ferromagnetism, the authors should do more investigations and experiments due to its lack of the 2D ferromagnetic property. Furthermore, 2D crystal should be perfectly manipulated (its thickness and orientation) as well.

Reply: Thanks reviewer for recognizing the importance and novelty of our work.

(1) Actually, there are four major notable points of about 2D CrTe crystal and its ferromagnetism in this work:

(I) Freestanding ultrathin non-van der Waals 2D CrTe crystals with thickness down to mono, few and multi-unit cell (UC) are acquired by a CVD-assisted ultrasonication method, showing great potential to prepare freestanding non-van der Waals intrinsic ferromagnetic 2D crystals, study their 2D quantum confinement effect and be applied in spintronics, magnetic resonance imaging, ferrofluid and drug carrier. Note that freestanding mono-UC non-van der Waals

2D CrTe crystals do exist and is obtained by this reported new method. See more details in notes for point 1 below.

(II) The Curie temperature (T_c) of freestanding 2D CrTe crystals is higher than bulk CrTe crystal, which is unexpectedly contrast with the discovered van der Waals intrinsic ferromagnetic 2D crystals which has much lower T_c than bulk crystal (Ref. 11-13, 15, Nature *546*, 265-269(2017); Nature *546*, 270-273(2017); Nat. Mater. *17*, 778-782(2018); Nat. Mater. *17*, 406-410(2018)), due to 2D quantum confinement effect induced enhancement of spin polarization with reduction of thickness and dimension. We also clearly show this unexpected opposite T_c behavior in freestanding non-van der Waals 2D crystals versus van der Waals ferromagnetic 2D crystals at single sheet level, consistent with the VSM results. The results herein open a new unexpected avenue and enable to achieve intrinsic, strong, room temperature or beyond room temperature 2D ferromagnetism in freestanding ultrathin 2D crystals for the first time, without any requirement for extrinsic modulations.

The mentioned papers are from the pioneering groups to realize the ferromagnetic 2D materials. In another recent works, the ferromagnetic phenomena are unveiled even in a few layer 2D materials (Nature Mater, *17*, 794 (2018), Sci Adv *6* eaay 8912 (2020), and so on), where the spin-orbit coupling is basically crucial to realize the 2D ferromagnetism and again it is originated from the well-defined atomic

structure. If this works want to provide the thickness effect, the authors should define the information on the exact thickness (not few layer, mult UC..) and the relevant clarification.

1. In Nature Communications, abstract is supposed to be within 150 words and unreferenced. The current abstract is now too long and more seriously it fails to emphasize the interesting issue in the current study.

Reply: Thanks for reviewer's comments. We have revised the abstract to the following:

“Ferromagnetism control is one of the most important fundamentals for modern electronics and spintronic, topological and quantum technologies in future. Inducing long-range ferromagnetic order in ultrathin 2D crystals will provide more functional possibility to combine their unique electronic, optical and mechanical properties to develop new multifunctional coupled applications. Recently discovered intrinsic 2D ferromagnetic crystals such as $\text{Cr}_2\text{Ge}_2\text{Te}_6$, CrI_3 and Fe_3GeTe_2 are intrinsically ferromagnetic only below room temperature, mostly far below room temperature (T_c , ~20-207 K). Here we develop a scalable method to prepare freestanding non-van der Waals ultrathin 2D crystals down to mono- and few unit cells (UC) and report unexpected strong, intrinsic, ambient-air-robust, room-temperature ferromagnetism with T_c up to ~367 K in freestanding non-van der Waals 2D CrTe crystals. Freestanding 2D CrTe crystals

show comparable or better ferromagnetic properties to widely-used Fe, Co, Ni and BaFe₁₂O₁₉, promising as new platforms for room-temperature intrinsically-ferromagnetic 2D crystals and integrated 2D devices.”

Accepted.

2. The terminology needs to be reconsidered. "Few-UC" and "Multi-UC" are not scientific at all. Together with this wrong terminology, the authors must define "the exact number of layers" because the quantum phenomena and magnetic properties can be drastically changed even by a single unit-cell thickness.

Reply: For a van der Waals crystal, since it is a layered structure, people are used to using "layer" to describe its thickness. But for non-van der Waals crystals, they do not have a layered structure, so "unit cell (UC)" is often used to evaluate their thickness (Nat. Mater. **9**, 397–402 (2010); Nat. Mater. **14**, 801–806 (2015); Phys. Rev. Lett. **103**, 146101 (2009); Phys. Rev. Lett. **102**, 176805 (2009); Phys. Rev. Lett. **119**, 256404 (2017); Sci. Rep. **4**, 6040 (2014)).

These works provided the exact number of unit cells to describe the thickness or superlattice layer. The main point in this paper is the thickness effect. The author should provide the exact number of unit cells from the measured sample. I do never understand the following description: "In freestanding form, the ultrasonically-splitted CrTe nanosheets must contain a certain thickness range. As shown below, the few-UC is defined as the thickness between 0.8 – 10 nm (1 - 12 UC), and mainly distributed in 2

- 4 nm (2 – 5 UC). The multi-UC is defined as the thickness between 10 - 60 nm (12 – 75 UC) and mainly distributed in 20 - 40 nm (25 – 50 UC). We have annotated their thickness in the discussions.”

The properties of 12 UC thick sample must be totally different to that of 75 UC thick sample. The multi-UC emerging the 12-75 UC cannot be accepted in the scientific reports.

We note that, contrast with van der Waals 2D ferromagnetic crystals, the sharp changes with thickness are not observed in the non-van der Waals CrTe 2D crystals.

Figure Rearranged from Fig. 2b and Supplementary Fig. 2b: Thickness distribution of few-UC (left) and multi-UC (right) CrTe crystals from AFM statistical analysis.

3. Figure 1g is HRTEM image from few UC sample. If its thickness is assumed to be within 3 layers, the HRTEM image is supposed to provide the phase information of Cr and Te atoms because the weak phase object approximation can be applied. That means there should be the ordered contrast pattern by Te atom with a higher atomic number. On the other hand, figure 1g shows no periodic pattern, which indicates that Cr and Te atoms are not crystallographically, atomically arrayed with a long-range order

parameter. If it is not ordered, the origin of the current work will falter. To this end, how about doing HAADF-STEM imaging or (X-ray or neutron) diffraction refinement?

Reply: As shown in Fig 1e-f or below, the selected area electron diffraction (SEAD) of CrTe nanosheet crystal is perfect hexagonal lattice arrangement, which indicates a well crystalline and uniformity of thickness of the CrTe nanosheet crystal. The reduced FFT image from the SEAD also indicates the high crystallinity of CrTe nanosheet 2D crystal. Our HRTEM also shows clear lattice fringes of (110) plane. But due to strong room temperature ferromagnetism influence of CrTe nanosheet 2D crystal on the electron focusing, we couldn't clearly see the atomic distribution to distinguish Cr and Te atoms.

Fig. 1e-f: **e**, HRTEM image of the few-UC CrTe crystal; **f**, The selected-area electron diffraction (SAED) for the few-UC CrTe crystal. **g**, Reduced Fast Fourier Transformation (RFFT) image derived from (**f**).

As I commented, the synthesized CrTe nanosheet is not atomically well defined just like in Figure 1, how can the authors be sure that the measure properties are genuine characteristic from the CrTe? Figure 2g must be a reverse Fast Fourier Transformed image from the diffraction pattern of Figure 2f, which can't imply anything at all.

Without the structural clarification, the authors' finding will lose the persuasive power.

4. How can the authors to normalize the magnetic moments (by emu/g) in cases of Multi-UC and Few-UC samples (Figure 2b~f) by dividing the amount of samples? The magnetic properties should be dealt in much more detail.

Reply: We've showed the details about normalizing the magnetic moments in our Supplementary Information. As mentioned in Characterizations and measurements in Supplementary Information, we drop a mass of 2D CrTe crystal solution onto the millipore filter substrate as Fig. 2c and Supplementary Fig. 5 show. The 2D CrTe crystals are easily oriented parallel to the substrate for their 2D ultrathin nature. The millipore filter substrate has been measured to be diamagnetic to preclude any influence and thus has no contribution to the ferromagnetism of the samples (Supplementary Fig. 6). To determine the mass of collected multi- and few-UC crystals, we weigh the millipore filter substrate before and after CrTe 2D crystals deposition with an electronic balance of high precision (0.01mg). To gain highly oriented CrTe crystals, the amount of the deposited 2D CrTe crystals are controlled in range of 2~3 mg. In order to get more accurate magnetization results, we prepare three different batches of samples for magnetization measurements.

Understood.

5. Because this paper is about ferromagnetic 2D materials, it is necessary to define the direction of magnetic field (i.e., along the in-plane direction or along the out-of-plane direction). There will be bunch of information about the 2D magnetic properties such

as the magnetic anisotropy and the subsequent spin Hall effect.

Reply: As mentioned above, we have shown the anisotropy of our few-UC and multi-UC CrTe crystals in Supplementary Fig. 8a, b, both of which show in-plane preferred easy-axis. Furthermore, we have also verified the in-plane easy-axis of a single sheet multi-UC CrTe crystal through magnetooptical Kerr rotation tests, as shown in Supplementary Fig. 8c, d below. Due to the in-plane preferred easy-axis of our CrTe crystals, all of other magnetic measurement were carried out with a-b plane parallel to the magnetic field (in-plane alignment). Due to the in-plane magnetization feature, anomalous Hall effect cannot be observed due to the Hall effect requires sample plane perpendicular to the magnetic field. However, although no anomalous Hall signal is detected by electrical method, we have detected the Kerr rotation signals by the magnetooptical method. As shown in Supplementary Fig. 8, the CrTe crystals show clear ferromagnetic Kerr rotation behavior with in-plane easy-axis.

Supplementary Fig. 8: Magnetic anisotropy of multi- and few-UC CrTe crystals.
a, b, The in-plane and out-of-plane M-H hysteresis loops for multi-UC (a) and few-UC

(b) CrTe crystals at 300 K. **c, d**, The out-of-plane (c) and in-plane (d) Magneto-optical Kerr rotation loops for a single sheet multi-UC CrTe crystal at 300 K.

Accepted.

6. The most serious point is that there's no indication of whether the magnetic property is intrinsic or extrinsic. Since a CrTe crystal is not a van der Waals crystal, it cannot be simply free of defects at the surface. In this CrTe crystal, the magnetic property can be changed by the type of termination layer (Cr termination, Te termination, half unit-cell termination, and so on..). If the authors want to insist on the intrinsic properties, the surface structure should be defined (for example, by STM).

Reply: (1) Regarding the question of whether it is intrinsically ferromagnetic, we would like to address this from the following 4 points. **First**, the bulk CrTe crystal is an intrinsic ferromagnetic material as reported and confirmed by VSM tests in Fig. 3a and Fig. 3d. **Second**, spontaneous magnetization regime tests without external magnetic field ($B=0$, zero magnetic field) of 2D CrTe crystals show clear transition at the Curie temperature, similar with FC-ZFC magnetization measurements, and further confirm its intrinsic feature (Fig. 3d). **Third**, The saturation magnetic moment of CrTe nanosheets obtained in our work is close to 100 emu/g, which is $10^5\sim 10^9$ times that of defect-induced magnetism (generally $\mu\text{emu/g}$ to memu/g , Appl. Phys. Lett. **104**, 202406(2014)), implying that our multi- and few-UC crystals are more like intrinsic ferromagnetism, not defects-induced ferromagnetism. **Finally**, the CrTe crystals have a magneto-optical Kerr rotation effects on polarized light, which also implies its intrinsic

ferromagnetic properties from optical perspective.

Fig. 3d Spontaneous magnetization regime tests without external magnetic field ($B=0$) for bulk, multi-UC and few-UC CrTe crystals from 2 K to 400 K.

(2) As for the effect of terminated atoms on ferromagnetism of CrTe crystals, we have revealed that the ferromagnetic moments of ultrathin 2D CrTe crystals mainly come from 3d orbital of Cr atom, as shown in Supplementary Fig. 13. Te atoms have negligible contribution to the ferromagnetism. From Supplementary Fig. 16, we can see that the magnetic moment of outer Cr atoms are slightly larger than inner Cr atoms. If the exposed surfaces are terminated by Cr atoms, we will expect slightly larger magnetic moment. Otherwise, for Te termination, the magnetic moment will slightly smaller than Cr- terminated one. Moreover, the VSM results show the saturated magnetic moment of few-UC CrTe 2D crystals up to 89 emu/g, $10^5\sim 10^9$ times larger than typical defect induced ferromagnetic moment, which cannot be dominated by termination-induced defect ferromagnetism.

Supplementary Fig. 13: The spin-resolved density of states (DOSs) and spin density distributions of bulk, multi-UC, few-UC and disordered few-UC CrTe crystals. a-d, Total DOS and partial DOS of Cr-3d states in bulk (a), multi-UC (b), few-UC (c) and disordered few-UC (d) CrTe crystals. The vertical dash lines denote the position of Fermi level. e, f, Spin density distribution of ordered few-UC (e) and disordered few-UC (f) CrTe crystal. The red and blue isosurfaces represent positive and negative spin density, respectively.

Supplementary Fig. 16: The calculated atomic magnetic moments of bulk (a), multi-UC (b) and few-UC (c) CrTe crystals.

(3) For the exposed surface, since the multi/few-UC crystals are a collection of ultrasonication-exfoliated nanosheets, which are obtained by splitting thick crystals along a specific crystal plane of lowest exfoliation energy, then there must be both Cr

atoms exposed surfaces in some sheets and Te atoms exposed surfaces in others. For the CrTe single crystals directly grown by CVD, the surface energy of the (002) plane by Te termination is the lowest according to our theoretical calculations as discussed below with more detailed analysis. We also note that it is very hard to get perfectly deposited samples from freestanding ultrathin sheets in solution for STM tests to identify the atoms of top layer surface.

To simulate the surface of NiAs-type CrTe, we adopt a slab model with 17 atomic layers and a vacuum region of 12Å to avoid interaction with each other (Fig. R1a,b). The (002) surface can be terminated either by Cr atomic layer or Te atomic layer (labeled as (002)-Cr and (002)-Te respectively). First, the topmost five layers are relaxed. Based on the relaxed structure, we calculated the surface energy to study the relative stability of the surface. The surface energy can be expressed as:

$$E^{surf} = \frac{1}{A} [E_{tot}^{slab} - N_{Te} E_{CrTe}^{bulk} + (N_{Te} - N_{Cr}) \mu_{Cr}],$$

Where A is the surface area, E_{tot}^{slab} is the total energy of the slab, E_{CrTe}^{bulk} is the energy of bulk CrTe, N_{Cr} and N_{Te} are numbers of Cr and Te atoms in the slab, and μ_{Cr} is the chemical potential of Cr. μ_{Cr} is related with the chemical potential of Te μ_{Te} with :

$$\mu_{Cr} + \mu_{Te} = \mu_{CrTe}^{bulk},$$

The Cr and Te chemical potential in the slab must be less than the corresponding elemental bulk chemical potential:

$$\mu_{Cr} \leq \mu_{Cr}^{bulk}, \mu_{Te} \leq \mu_{Te}^{bulk}.$$

The variation of μ_{Cr} is:

$$\mu_{CrTe}^{bulk} - \mu_{Te}^{bulk} \leq \mu_{Cr} \leq \mu_{Cr}^{bulk}$$

The result shows that the Te-terminated (002) surface has lower surface energy than Cr-terminated (002) surface over the effective chemical potential range, hence the Te-terminated surface are energetically more stable (Fig. R1c).

Fig. R1: The slab model for the surface structure of NiAs-type CrTe and corresponding surface energies. a, The Cr-terminated (002) surface. **b,** The Te-terminated (002) surface. **c,** Surface energies as a function of chemical potential of Cr for the Cr and Te terminated (002) surface.

Thanks for the hard work.

7. Figures 1~3 are all not clear and thus the author cannot represent the meaning of the results. There are too many sub-figures but there are no link between them in each figure.

I ask the authors to choose the most important data which are fitting into the main concept of each figure. FYI, the font sizes are too small to recognize some of them.

Reply: Some displayed figures may be too small to be effectively recognized as reviewer said. We have put unimportant figures in the supplementary information and

enlarged the remained figures. As for Fig. 1, we mainly intend to show the crystal structure of non-van der Waals CrTe, indicating that the (002) plane is easy to break, and show the ultrathin 2D CrTe crystal obtained by CVD-assisted ultrasonication method. Fig.2 shows characterizations for few-UC CrTe crystals. Fig. 3 shows the ferromagnetism of three kinds of thicknesses of CrTe crystals and their comparison with traditional ferromagnetic materials. Fig. 4 shows room temperature ferromagnetism of few-UC CrTe crystals by single sheet MFM tests at nanoscale.

Look more understandable now.

8. I don't think figure 4 is quite necessary in the main figure set at the moment. The author may move it into the supporting information. In my opinion, the authors must more concentrate on unveiling (1) whether it is intrinsic or extrinsic and (2) the typical, representative magnetic properties in the field of 2D ferromagnetism.

Reply: Figure 4 in original manuscript mainly talks about the relatively stable ferromagnetic moment of the multi- and few-UC CrTe crystals under the ambient environment, contrast with the instability of most van der Waals 2D ferromagnetic crystals in air. As suggested, we have remove this part to the supplementary information. Regarding whether the magnetic moment of CrTe is intrinsic or extrinsic, all evidence indicate an intrinsic ferromagnetism as discussed in Reply to Question 6. Its magnitude of saturated magnetic moment, the spontaneous magnetization without external magnetic field ($B=0$), and magneto-optical Kerr loops all indicate that it is intrinsically

ferromagnetic rather than extrinsically.

Now the figure set looks more consistent.

Reviewer #2:

Remarks to the Author:

In this revised manuscript and reply, the author address the importance of their research for the non van Der Waals magnetic crystals, towards the application based on their current forms(crystals but not thin films). I was kindly convinced by their argument.

Additionally, the author added new results on the LDW effect observed by the Kerr measurement, saying LDW only is observed on ultra-thin CrTe crystals, and random domain walls are observed on the thick crystal. However, by comparing the figS10 and FigS11, I didn't get the clear idea for the RDW effect in the thick sample. In fact, I do observed a linear-like fringe in the FigS11. I think the author should give more careful analysis on the domain wall behaviors on both the thin and thick CrTe sample, for making their claim convincing.

Response to reviewers

Reviewer #1:

Thanks for the great efforts to enhance the paper.

However, the crystallographic or atomic-structural understanding is still ambiguous.

And there still remain the thickness issues as well. Please see the attached file.

Reply: Thanks for the reviewer's valuable advice. Regarding the atomic-structure of the crystal, we carefully adjusted the Gun Deflector and Condenser Stigmator to make the obtained electron beam spot on the sample as uniform as possible and we have obtained clearer HRTEM atomic images of ultrathin CrTe crystals. The structural model of CrTe crystal unit cell is shown in Response Figure 1a, and Cr and Te atoms are hexagonally distributed as viewed along [002] direction (Response Figure 1b). Response Figure 1d shows the atomic-resolution TEM image of the CrTe ultrathin crystal, which presents a clear hexagonal structure and is fully consistent with the cell structure of CrTe crystal when viewed at [002] direction (Response Figure 1b). The obtained atomic structure is also fully consistent with the calculated atomic structures from reduced fast Fourier transformation (RFFT) of electron diffraction patterns (Response Figure 1e, f). Combining the structural models and atomic structures from newly-added HRTEM characterizations with the XRD, XRF and EDS spectra, the crystal structure of CrTe ultrathin crystal is more clearly described in this revision.

Response Figure 1. HRTEM images of CrTe thin crystals. (a) Unit cell structure of CrTe crystal. (b) Surface structure of (002) facet for CrTe crystal. (c) HRTEM image at the edge of the few-UC CrTe crystal (Newly added figure, which also can be seen in maintext Fig. 2e). (d) **Atomic-resolution TEM image of the few-UC CrTe crystal (Newly added figure, which also can be seen in maintext Fig. 2f).** (e) The selected-area electron diffraction (SAED) for the few-UC CrTe crystal. (f) Reduced Fast Fourier Transformation (RFFT) image derived from (e).

As for the correspondence between the exact thickness and related physical properties, we have further supplemented the anomalous Hall effect (AHE) and the magnetic force microscope (MFM) tests for single-sheet CrTe crystals of different exact thickness (Response Figures 2-5). For the AHE test, five-wire Hall device was employed to offset the impact of R_{xx} component on R_{xy} , as shown in

Response Figure 2c. Finally, we have obtained the temperature-dependent Hall resistivity for four single-sheet CrTe crystals of different thicknesses (100 nm, 69 nm, 37 nm thin crystals and 20 μm bulk crystal), as shown in Response Figure 3 and Response Figure 4. The Curie temperatures from AHE for 100 nm, 69 nm, and 37 nm CrTe Crystals are ~ 340 K, 370 K, and 360 K, respectively, all higher than the 330 K Curie temperature from AHE for the bulk CrTe crystal. T_c increases first and then decreases with the decrease of thickness, which is consistent with the previous VSM and Kerr results. At room temperature, the anomalous Hall resistivities of bulk, 100 nm, 69 nm and 37 nm CrTe crystals are 0.64, 0.82, 12.7 and 1.95 $\mu\Omega\cdot\text{cm}$, respectively. The anomalous Hall resistivity also increases first and then decreases with the decrease of thickness, which is also consistent with the change trend of T_c . Unfortunately, CrTe crystals below 10 nm thick is not possible for AHE tests because the small crystal size. However, from the above tested several single sheet CrTe crystals of exact thickness, it is clear that the T_c by AHE increases first and then decreases with the decrease of thickness, which is consistent with our previous VSM and magnetooptical Kerr tests and is contrast from the van der Waals ferromagnetic crystals of T_c always decreasing with thickness reduction.

In addition to the change of T_c , we also investigated the variation of magnetic domains with thickness of CrTe crystals through magnetic force microscope (MFM) at room temperature. When the CrTe crystal is in bulk form (~ 20 μm), the shape of the magnetic domain wall is arranged randomly under the zero magnetic field, as shown in Response

Figure 5a, b. When the thickness of CrTe is reduced to several hundreds of nanometers (673 nm), the shape of the magnetic domains is a mixture of random and striped magnetic domains. When the thickness is reduced to 96, 39 and 34 nm, only linear magnetic domains are found. When CrTe crystals were ultrasonically exfoliated to several nanometers thick, due to the much thinner thickness of the nanosheets, there is only one magnetic domain on a single sheet, whose shape is exactly the same with the CrTe crystal, as shown in Figure 4 in maintext.

Response Figure 2 (Newly added figure, which also can be seen in Supplementary Fig. 11). (a, b) Optical photographs of the as-grown CrTe single-nanosheet crystals with a size of several hundred microns; (c) Schematic diagram of five-wire Hall device. The inset is a photo of the Hall device made by dipping 25 μm gold wire in silver glue and pressing onto the sample.

Response Figure 3 (Newly added figure, which also can be seen in Supplementary Fig. 12). Anomalous Hall effect of single-sheet CrTe crystals with different thickness. a-c, AFM morphologies of CrTe crystals with three different thicknesses, the inset shows the corresponding thickness profiles; d-f, Anomalous Hall effect of 100 nm (d), 69 nm (e) and 37 nm (f) CrTe crystals in (a-c).

Response Figure 4 (Newly added figure, which also can be seen in Supplementary Fig. 13). Anomalous Hall resistivity of bulk CrTe. The thickness is determined to be ~20 μm by a micrometer.

Response Figure 5 (Newly added figure, which also can be seen in Supplementary Fig. 16). Room temperature MFM magnetic domain imaging of CrTe single crystals with different thicknesses. (a, b) MFM phase image (a) and corresponding 3D view (b) of a 20 μm thick CrTe crystal. (c-e) AFM image (c), MFM phase image (d) and corresponding 3D view (e) of a 673 nm thick CrTe crystal. (f, g) AFM image (f) and MFM phase image (g) of CrTe crystals with thickness of 34, 39 and 96 nm.

Fig. 4 in maintext: Direct imaging the room temperature ferromagnetic properties of single sheet few-UC CrTe crystals by magnetic force microscopy (MFM) at nanoscale.

Reviewer #2:

In this revised manuscript and reply, the author address the importance of their research for the non van Der Waals magnetic crystals, towards the application based on their current forms(crystals but not thin films). I was kindly convinced by their argument.

Reply: Thank the reviewer for the recognition of the importance of our work. Based on the reviewer's advice, we carefully marked the structure of the magnetic domains in the

Kerr image and further clarified the law of magnetic domain changes with thickness by Magnetic Force Microscopy (MFM) in this second revision. The results are consistent with the previous Kerr test, which further confirms the previous results.

Additionally, the author added new results on the LDW effect observed by the Kerr measurement, saying LDW only is observed on ultra-thin CrTe crystals, and random domain walls are observed on the thick crystal. However, by comparing the figS10 and FigS11, I didn't get the clear idea for the RDW effect in the thick sample. In fact, I do observed a linear-like fringe in the FigS11. I think the author should give more careful analysis on the domain wall behaviors on both the thin and thick CrTe sample, for making their claim convincing.

Reply: In order to show the shape of the magnetic domain walls in the bulk crystal more clearly, we have carefully adjusted the contrast of the picture, and the magnetic domain wall is marked in green, as shown in the figure below. Obviously, all the magnetic domain walls in the bulk crystal are arranged randomly. The linear-like fringe that the reviewer noticed is reflective stripe caused by topography, not the magnetic domain wall. The magnetic domain wall separates two magnetic domains with different brightness, but the brightness of the magnetic domains on both sides of this linear-like fringe is exactly the same.

Newly Processed Supplementary Fig. 15 Room temperature magnetooptical tests of a single sheet bulk CrTe crystal under magnetic fields of 5 mT (a) and 15 mT (b), showing random domain walls and no LDW found. Thickness of bulk CrTe crystal is $\sim 5.1 \mu\text{m}$ (~ 6375 UC). Note that the bright line in the red dotted circle is the high reflection line caused by the morphology of the sample, not the domain wall. The green solid lines represent random magnetic domain walls, that separate different domains. (d, e) The fine domain structure on sample (d, blue box in a) and substrate (e, yellow box in a).

To further prove the thickness dependence magnetic domain wall behavior of the CrTe single crystals. MFM measurements were carried out on CrTe crystals with different thickness of 20 μm , 673 nm, 96 nm 39 nm and 34 nm (Response Figure 5). When the thickness of the sample is 20 μm (bulk CrTe), the shape of the magnetic domain is hilly and undulating, and the shape of the magnetic domain wall is arranged randomly,

agreeing with previous Kerr measurements. When the thickness of CrTe is reduced to several hundreds of nanometers (673 nm), the shape of the magnetic domains is a mixture of random and striped magnetic domains. Therefore, the shape of the magnetic domain wall is also composed of random magnetic domain walls and linear magnetic domain walls. When the thickness is further reduced to 96 nm, 39 nm and 34 nm, only linear magnetic domains are found, which is in good agreement with the results of the previous Kerr measurements. It is worth mentioning that when CrTe crystals were ultrasonically exfoliated to several nanometers thick, due to the much thinner thickness of the nanosheets, there is only one magnetic domain on a single sheet, whose shape is exactly the same with the CrTe crystal, as shown in Figure 4 in maintext.

Response Figure 5 (Newly added figure, which also can be seen in Supplementary Fig. 16). Room temperature MFM magnetic domain imaging of CrTe single crystals with different thicknesses. (a, b) MFM phase image (a) and corresponding 3D view (b) of a 20 μm thick CrTe crystal; (c-e) AFM image (c), MFM phase image (d) and corresponding 3D view (e) of a 673 nm thick CrTe crystal; (f, g) AFM image (f) and MFM phase image (g) of CrTe crystals with thickness of 34, 39 and 96 nm.

Fig. 4 in maintext: Direct imaging the room temperature ferromagnetic properties of single sheet few-UC CrTe crystals by magnetic force microscopy (MFM) at nanoscale.

Reviewers' Comments:

Reviewer #1:

Remarks to the Author:

I am sure that the current version is much improved than before. Thankful to all of authors for the sincere response and contribution to Nature communications. I think the paper has become acceptable.